# *Porphyromonas gingivalis* aggravates colitis via a gut microbiota-linoleic acid metabolism-Th17/Treg cell balance axis

Lu Jia [1], Yiyang Jiang[1], Lili Wu[1], Jingfei Fu[1], Juan Du[1], Zhenhua Luo[1], Lijia Guo[2], Junji Xu[1]✉ & Yi Liu [1]✉

Periodontitis is closely related to inflammatory bowel disease (IBD). An excessive and non-self-limiting immune response to the dysbiotic microbiome characterizes the two. However, the underlying mechanisms that overlap still need to be clarified. We demonstrate that the critical periodontal pathogen *Porphyromonas gingivalis* (Pg) aggravates intestinal inflammation and Th17/ Treg cell imbalance in a gut microbiota-dependent manner. Specifically, metagenomic and metabolomic analyses shows that oral administration of Pg increases levels of the *Bacteroides* phylum but decreases levels of the *Firmicutes*, *Verrucomicrobia*, and *Actinobacteria* phyla. Nevertheless, it suppresses the linoleic acid (LA) pathway in the gut microbiota, which was the target metabolite that determines the degree of inflammation and functions as an aryl hydrocarbon receptor (AHR) ligand to suppress Th17 differentiation while promoting Treg cell differentiation via the phosphorylation of Stat1 at Ser727. Therapeutically restoring LA levels in colitis mice challenged with Pg exerts anti-colitis effects by decreasing the Th17/Treg cell ratio in an AHR-dependent manner. Our study suggests that Pg aggravates colitis via a gut microbiota-LA metabolism-Th17/Treg cell balance axis, providing a potential therapeutically modifiable target for IBD patients with periodontitis.

Periodontitis is an inflammatory response triggered by pathogenic microorganisms in the oral cavity and is the leading cause of tooth loss in humans. Periodontitis patients may swallow $10^1$-$10^{13}$ bacteria in saliva every day[1]. Among them, the gram-negative anaerobic microbe *Porphyromonas gingivalis* (Pg) functions as a keystone pathogen and possesses an array of virulence factors[2]. Periodontitis has a major impact on various systemic diseases, including inflammatory bowel disease (IBD)[3–5], Type 2 diabetes mellitus (T2D), metabolic syndrome, and coronary heart disease[6]. The intrinsic mechanisms involve distant spread and ectopic infection by periodontopathogens, distal end migration of oral inflammatory-activated Th17 cells, and metabolic and immune disorders[7]. IBD is a specific type of intestinal inflammation and is classified into two major forms, Crohn's disease (CD) and

ulcerative colitis (UC). There is substantial evidence for the association between IBD and periodontitis[8,9]. In fact, both diseases are characterized by an excessive and nonself-limiting immune response to the dysbiotic microbiome[10]. However, the underlying mechanisms that overlap between the conditions remain unclear.

T helper 17 cells (Th17) are a subset of T cells that mainly secrete proinflammatory factors such as IL-17 and IL-22 and differentiate from naïve T cells under the stimulation of IL-6 and TGF-β. Retinoic acid orphan receptor γ (RORγ) is the transcription factor preferentially expressed in Th17 cells and is essential for their differentiation and development[11,12]. Th17 cells are involved in the pathogenesis of the most common autoimmune diseases, including IBD, rheumatoid arthritis (RA), and multiple sclerosis (MS). Recently, researchers have

[1]Laboratory of Tissue Regeneration and Immunology and Department of Periodontics, Beijing Key Laboratory of Tooth Regeneration and Function Reconstruction, School of Stomatology, Capital Medical University, Beijing, P. R. China. [2]Department of Orthodontics School of Stomatology, Capital Medical University, Beijing, P. R. China. ✉e-mail: uujkl@163.com; liliiuyi@163.com

found that Th17 cells are at the core of IBD and are related not only to the mucosal barrier but also to inflammation[13]. Regulatory T cells (Treg cells) are anti-inflammatory regulators that produce IL-10, TGF-β, and other cytokines under TGF-β stimulation. The developmental pathways of Th17 and Treg cells are closely related; that is, they regulate and differentiate each other to maintain the Th17/Treg cell balance, thereby affecting the outcome of IBD[14].

Gut microbiota dysbiosis is increasingly considered a core pathogenic mechanism underlying many diseases. Gut dysbiosis in patients with IBD has been well documented[15]. Periodontal pathogens can affect gut microbiota dysbiosis through ectopic colonization, pathological factors, or virulence factors[16]. Recently, scholars have introduced the concept of the "gum-gut" axis as a framework for examining the reciprocal relationship between the periodontium and the gastrointestinal tract[17]. Gut microbiota disturbance is an important process through which periodontitis affects IBD, but the detailed mechanism remains to be elucidated. As byproducts of the gut microbiota, metabolites are vital factors in host-microbiota crosstalk. Accumulating evidence indicates that the complex interaction between metabolism and immunity is the essential mechanism regulating homeostasis in IBD[18,19]. Short-chain fatty acids (SCFAs) promote the production of IL-22 by CD4+ T cells during colitis, which is mediated by the induction of butyrate production via the AHR, which is a function of gut microbiota-derived metabolites that regulates intestinal homeostasis[20]. A systematic review examined more than 40 metabolites as periodontitis-specific biomarkers, mainly related to amino acid and lipid degradation pathways[21], and oral administration of Pg aggravated metabolic disorders; it promoted a higher proportion of fat mass, worse glucose tolerance, and lipid/amino acid metabolic alteration[22].

To better understand how periodontitis affects colitis via the "gum-gut" axis, we investigate how intragastric administration of Pg affects intestinal inflammation. To identify the role of the gut microbiota and metabolic regulation in the effects of Pg gavage on dextran sodium sulfate (DSS)-induced colitis, gut microbiota depletion and faecal microbiota transplantation (FMT) combined with metagenomic and metabolomic sequencing experiments are conducted. Here, we show that gavage of Pg exacerbates colitis and Th17/Treg imbalance in a gut microbiota-dependent manner. Oral administration of Pg leads to increased *Bacteroidetes* phylum and a decreased proportion of *Firmicutes* phylum, which disfavours LA production in the gut. Mechanically, LA activates the phosphorylation of Stat1 (Ser727) via AHR to repress Th17 differentiation but promote Treg cell development.

## Results

### Gavage of Pg aggravated colitis

To determine whether periodontitis can affect colitis, we administered Pg to DSS-induced colitis mice. Wild-type (WT) mice were gavaged with Pg for 14 days and with 3.0% DSS during the last 7 days of treatment (Fig. 1a). Oral administration of live Pg (DSS+LiPg group) or Pg extract (DSS+DePg group) aggravated colitis. DSS+LiPg mice developed more severe inflammation than DSS+DePg mice based on body weight, disease activity index (DAI), colon length, histological activity index (HAI) (Fig. 1b–g) and cytokine expression (Supplementary Fig. 2a–d), suggesting that the degree of colonic inflammation was closely related to Pg activity. Next the percentages of IL-17+Th17 and Foxp3+Treg cell in mesenteric lymph nodes (MLNs) and lamina propria lymphocytes (LPLs) of colon were detected; the percentage of Th17 cells in the DSS+LiPg group and DSS+DePg group was significantly higher, and the percentage of Treg cells in the DSS+LiPg group and DSS+DePg group was significantly lower than that in the DSS + PBS group (Fig. 1h–m). RT–qPCR results further confirmed the expression of nuclear transcription factors (*Rorc*, *Foxp3*, *Stat3*, *Stat5*) in colon

homogenates, which also verified the above results (Supplementary Fig. 2e). The Th17/Treg cell ratios in the MLN and LPL in the DSS+LiPg group and DSS+DePg group were significantly higher than those in the control group (Fig. 1n), indicating that the aggravation of colitis by gavage with Pg was well reflected by the Th17/Treg cell imbalance. These findings indicated that Pg treatment significantly aggravated DSS-induced colitis and caused a Th17/Treg cell immune imbalance.

### Gavage of Pg exacerbated colitis and Th17/Treg cell imbalance in a gut microbiota-dependent manner

To determine whether the aggravation of colitis by gavage with Pg is due to the toxicity of Pg itself or the regulation of gut microbes, we first designed a gut microbiota clearance experiment. WT mice were administered quadruple antibiotic cocktails by gavage for gut microbiota depletion before DSS treatment (ABX) (Fig. 2a and Supplementary Fig. 1a). Then, Pg or Pg extract was administered to these colitis mice. Colitis in neither the ABX (DSS+LiPg) group nor the ABX (DSS+DePg) group was aggravated compared with that in the ABX (DSS + PBS) group (Supplementary Fig. 1b–g and Supplementary Fig. 2f–h, *p* > 0.05). There were no statistically significant differences in the percentages of Th17 and Treg cells or the Th17/Treg cell ratio in MLNs and LPLs among the ABX (DSS + PBS) group, ABX (DSS+LiPg) group and ABX (DSS+DePg) group (Supplementary Fig. 1h–n). The above results indicated that the aggravation of colonic inflammation and Th17/Treg cell imbalance caused by gavage of Pg to colitis mice were not directly triggered by Pg but might involve the gut microbiota.

To further confirm this possibility, we conducted an FMT experiment in which gut microbiota-depleted recipient mice (GDR mice) were reconstituted with the gut microbiota of DSS + PBS-treated mice (FMT (DSS + PBS) group), DSS+LiPg-treated mice (FMT (DSS +LiPg) group) or DSS+DePg-treated mice (FMT (DSS+DePg) group) via intragastric administration once a day for 5 days (Fig. 2a). The reconstruction of donor microbiota was confirmed by using metagenomic sequencing (Supplementary Fig. 3). The FMT (DSS+LiPg) group and FMT (DSS+DePg) group mice developed more severe inflammation than the FMT (DSS + PBS) group mice, as judged by weight loss, DAI, colon length, HAI (Fig. 2b–g) and cytokine expression (Supplementary Fig. 2i, j). Moreover, FMT (DSS+LiPg) mice developed more severe colitis than FMT (DSS+DePg) mice (Fig. 2b–g), suggesting that the enhanced colitis in GDR mice was caused by changes in the gut microbiota through the intake of faeces from donor mice. The percentage of Th17 cells and the Th17/Treg cell ratio in the FMT (DSS +LiPg) group and FMT (DSS+DePg) group were significantly higher than those in the FMT (DSS + PBS) group, and the percentage of Treg cells in the FMT (DSS+LiPg) group and FMT (DSS+DePg) group was significantly lower (Fig. 2h–n), indicating that the local Th17/Treg cell imbalance in the colon of recipient mice was dependent on the gut microbiota of the donor mice.

Considering the clearance of the gut microbiota using antibiotic cocktails prevented the pathological changes caused by administration of Pg, we determined whether residual antibiotics have an impact on the live Pg numbers in the intestine (Supplementary Fig. 4a, b). The Pg content in the feces of mice after gavage of Pg reached the peak on the 3rd day (Supplementary Fig. 4b, Day 3, $2.14 \times 10^9 \pm 3.29 \times 10^8$) and began to decrease from the 5th day (Day 5, $1.27 \times 10^8 \pm 2.28 \times 10^7$). The Pg content in the feces of mice after administration of antibiotics remained a very low level (Supplementary Fig. 4b, Day 5, $3.59 \times 10^7 \pm 1.75 \times 10^7$; Day 7, $2.29 \times 10^8 \pm 6.14 \times 10^7$). Since donor feces contained some Pg, the FMT experiment did not exclude the possibility that Pg directly influenced the inflammatory status and T cell balance, we conducted the experiments to support that the severity of colitis was due to the altered gut microbiota composition instead of the power of Pg itself (Supplementary Fig. 4c–i). To confirm that the

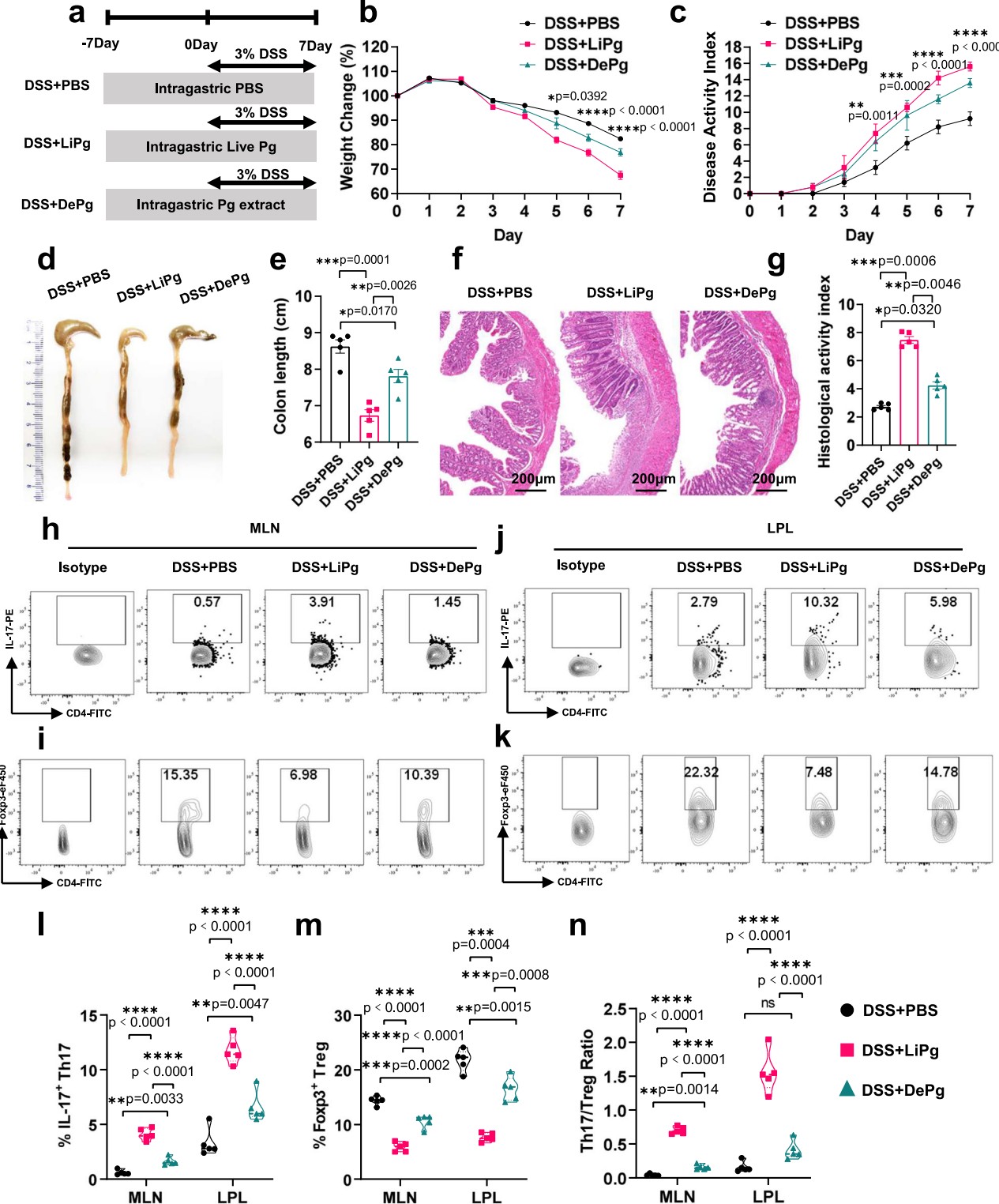

**Fig. 1 | Oral administration of Pg aggravated colitis. a** The experimental design. Pg is the key pathogen of periodontitis. A DSS-induced colitis model was established. Colitis mice were gavaged with live Pg (DSS+LiPg) or ultrasonic fragmentation extract of Pg (DSS+DePg). **b** Body weight of DSS + PBS, DSS+LiPg, and DSS +DePg mice during DSS-induced colitis, $n = 5$ biologically independent samples. **c**–**g** Disease activity index (DAI) (**c**), colon length (**d**, **e**), and histological score (**f**, **g**) of DSS + PBS, DSS + LiPg, and DSS + DePg mice on day 8 after DSS induction, $n = 5$ biologically independent samples. Scale bar, 200 μm. **h**–**k** The proportions of IL-17[+]

Th17 cells and Foxp3[+] Treg cells and their ratio among total CD4[+] T cells within the MLNs and LPLs were detected by flow cytometry and statistically analysed (**l**–**n**), $n = 5$ biologically independent samples. Data are presented as the mean ± SEM; *$p < 0.05$; **$p < 0.01$; ***$p < 0.001$ by two-tailed one-way ANOVA. Pg *Porphyromonas gingivalis*, MLN mesenteric lymph node cells, LPL colon lamina propria lymphocytes, ANOVA analysis of variance, DSS dextran sodium sulfate. Source data are provided as a Source Data file.

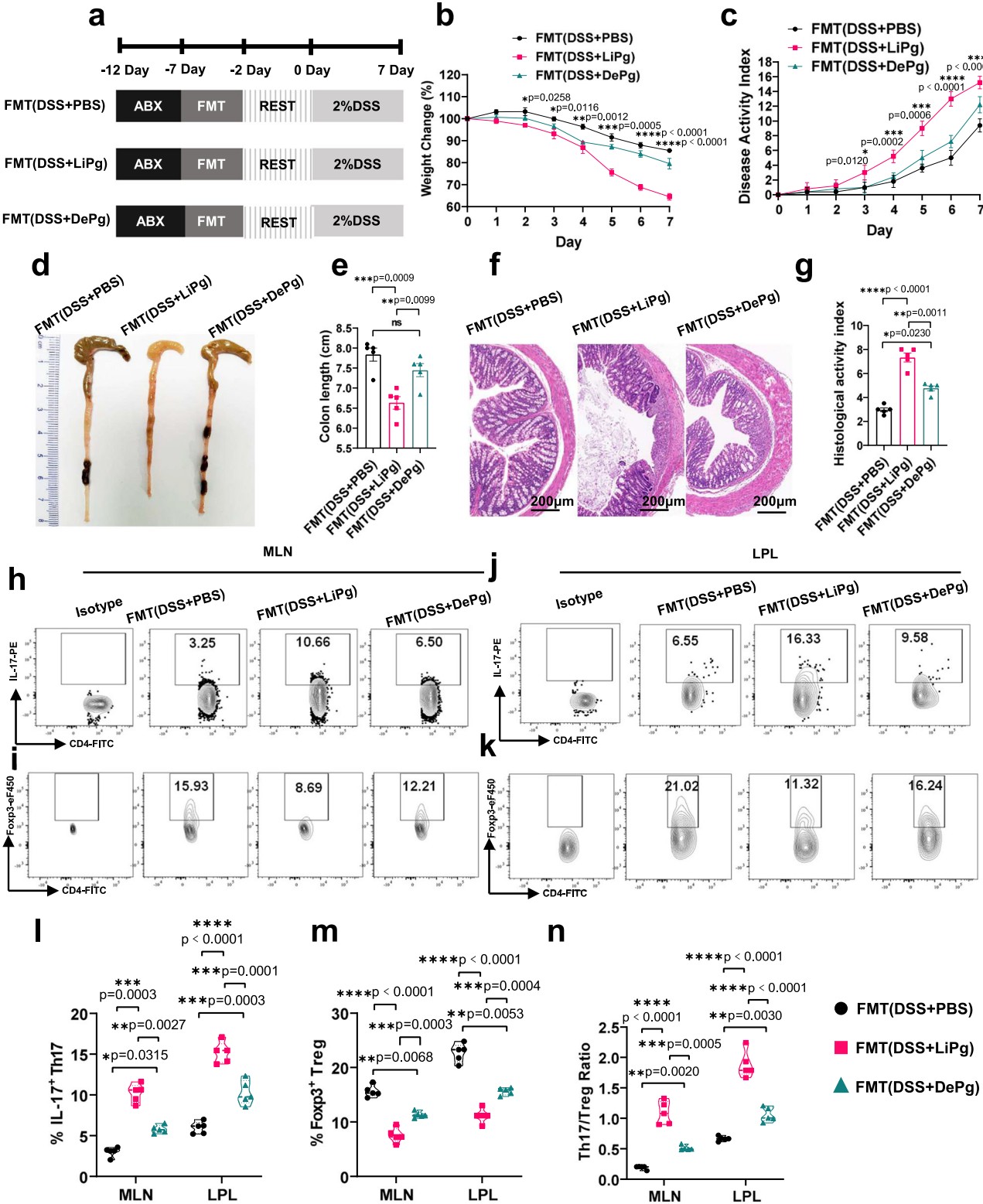

**Fig. 2 | Oral administration of Pg exacerbated colonic Th17/Treg cell imbalance in a gut microbiota-dependent manner. a** Diagram of the gut microbiota transplantation experiment. ABX: antibiotic cocktail. FMT: faecal microorganism transplantation. Transplantation of mouse faeces from the DSS + PBS/LiPg/DePg group into recipient mice→the FMT (DSS + PBS/LiPg/DePg) group. **b** Body weight of FMT (DSS + PBS), FMT (DSS+LiPg), and FMT (DSS+DePg) mice during DSS-induced colitis, *n* = 5 biologically independent samples. **c–g** Disease activity index (DAI) (**c**), colon length (**d, e**), and histological score (**f, g**) of FMT (DSS + PBS), FMT (DSS +LiPg), and FMT (DSS+DePg) mice on day 8 after DSS induction, *n* = 5. Scale bar,

200 μm. **h–k** The proportions of IL-17⁺ Th17 cells and Foxp3⁺ Treg cells and their ratio among total CD4⁺ T cells within the MLNs and LPLs were detected by flow cytometry and statistically analysed (**l–n**), *n* = 5 biologically independent samples. Data are presented as the mean ± SEM; **p* < 0.05; ***p* < 0.01; ****p* < 0.001; *****p* < 0.0001 by two-tailed one-way ANOVA. Pg *Porphyromonas gingivalis,* MLN mesenteric lymph node cells, LPL colon lamina propria lymphocytes, ANOVA analysis of variance, DSS dextran sodium sulfate. Source data are provided as a Source Data file.

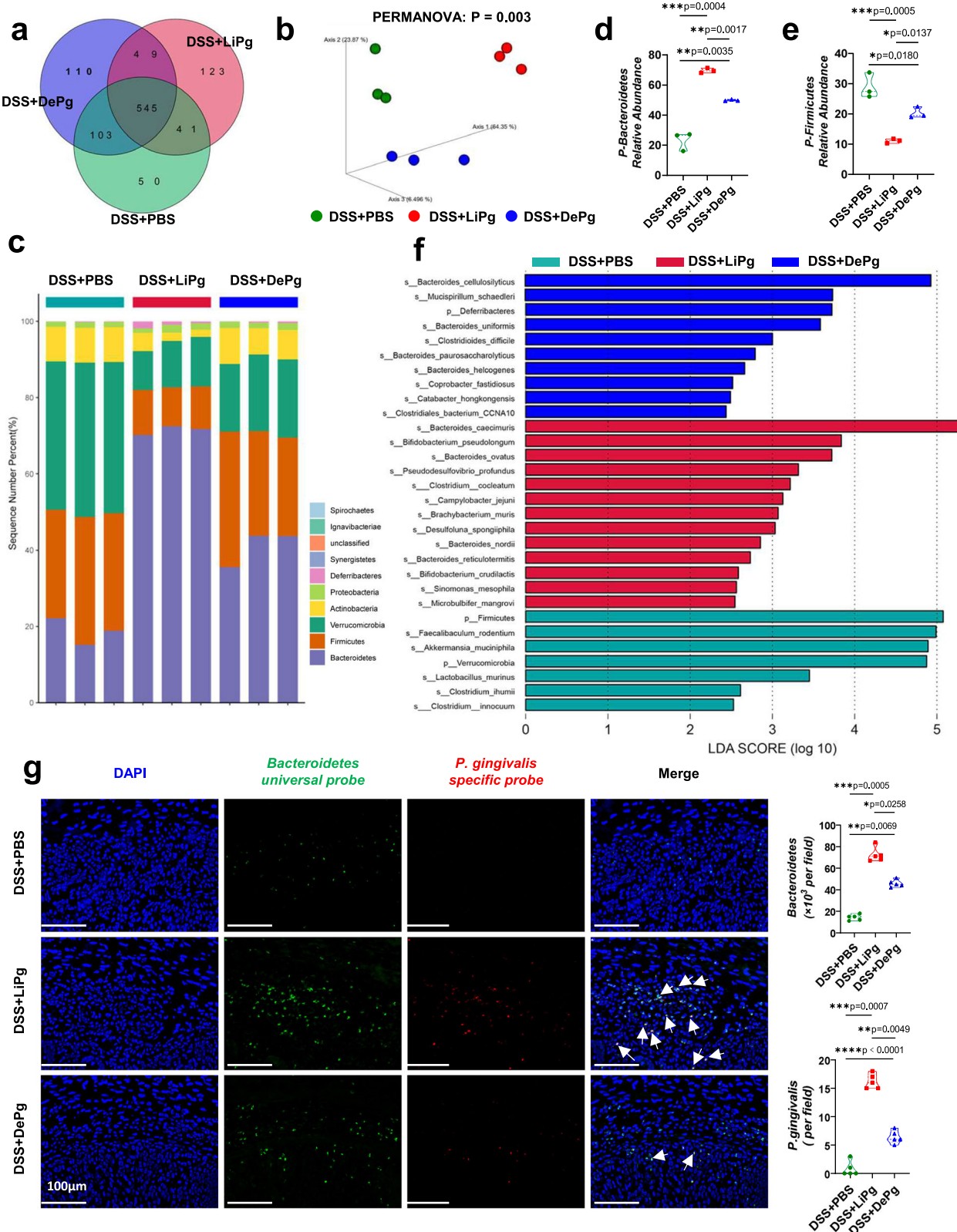

**Fig.** *(panels a–g)*

Th17/Treg cell imbalance was the cause, not the consequence of the exacerbation of colitis after gavage of Pg in a gut microbiota-dependent manner, we found that the increased Th17/Treg cell ratio also happened in the colon at the early stage of colitis (Day 2) after gavage of Pg (Supplementary Fig. 5). Together, our data revealed that the enhanced colonic susceptibility and Th17/Treg cell imbalance after gavage with Pg were related to changes in the gut microbiota.

**Gavage of Pg altered the gut microbiota composition**

To identify changes in the gut microbes of colitis mice after gavage of Pg, we performed high-throughput metagenomic sequencing of faecal bacterial genomes isolated from the DSS + PBS group, DSS+LiPg group and DSS+DePg group. Common species analysis (Fig. 3a) showed that the number of common species between the DSS+LiPg and DSS + PBS groups was significantly less than that between the DSS+DePg and

**Fig. 3 | Oral administration of Pg altered the gut microbiota composition.**
**a** High-throughput metagenomic sequencing of the faecal bacterial genome from
DSS + PBS, DSS+LiPg, and DSS+DePg mice. *n* = 3. The number of common or unique
species among different groups in the Venn diagram. **b** 3D PCoA graph based on the
Bray-Curtis distance matrix. PERMANOVA: F = 20.208, two-sided *P* value = 0.003.
PERMANOVA: permutational multivariate analysis of variance. **c** Column diagram of
the relative distribution of each sample at the phylum level (top 10). The *Y*-axis is
the sequence number percent, indicating the ratio of this phylum level to the total
annotation data. **d**, **e** The box plots indicate the top 2 relative abundances of each

bacterial group at the phylum level (*Bacteroidetes* and *Firmicutes*). **f** The most dif-
ferentially abundant taxa of characteristic microorganisms from DSS + PBS, DSS +
LiPg, and DSS + DePg mice (LDA Sore = 4 by LEfSe). LDA Score: linear discriminant
analysis score. **g** FISH of colonic lumen sections from DSS + PBS, DSS + LiPg, and
DSS + DePg mice: *Bacteroidetes* universal probe (green), *P. gingivalis*-specific probe
(red), and DAPI (blue). Scale bar, 100 μm. At least five sections per group were
analysed. Data are presented as the mean ± SEM; *$p < 0.05$; **$p < 0.01$; ***$p < 0.001$;
****$p < 0.0001$ by two-tailed one-way ANOVA with two-sided *p* value. Source data
are provided as a Source Data file.

DSS + PBS groups, indicating that the changes in the gut microbiota
after gavage of live Pg were more substantial than those after gavage of
Pg extract (no bacterial activity), which was consistent with the pre-
sence of more severe colonic inflammation in the DSS+LiPg group. The
Chao1 index of the alfa diversity analysis calculating the community
richness showed no significant difference among the three groups
(Supplementary Fig. 6a–c, p å 0.05). The Simpson indexes of the DSS
+LiPg group and DSS+DePg group were higher than that of the DSS +
PBS group, implying higher diversity after Pg administration (Supple-
mentary Fig. 6d–f, *p* = 0.027). The PCoA analysis based on the Bray-
Curtis distance showed that the samples from the DSS+LiPg group had
lower similarity than those from the DSS+DePg group in terms of
species composition structure compared with those from the DSS +
PBS group (Fig. 3b and Supplementary Fig. 6g). Species composition
analysis (Fig. 3c) at the phylum level showed that the top 4 bacterial
phyla with the highest abundance in all samples were *Bacteroidetes*,
*Firmicutes*, *Verrucomicrobia*, and *Actinobacteria*. Among these, the
abundance of *Bacteroidetes* was increased while the abundances of
*Firmicutes*, *Verrucomicrobia* and *Actinobacteria* were decreased in DSS
+LiPg mice (Fig. 3d, e and Supplementary Fig. 7a, b). The analysis of
species with significant differences based on LEfSe (Fig. 3f and Sup-
plementary Fig. 7d) showed that when LDA > 2, there were 13 bacterial
species in the DSS+LiPg group, 9 bacterial species in the DSS+DePg
group and 5 bacterial species in the DSS + PBS group with significant
differences. When LDA > 4, the microbes with a significant difference
in the DSS+LiPg group was *Bacteroides caecimuris*; that in the DSS
+DePg group was *Bacteroides cellulosilyticus*; and those in the DSS +
PBS group were *Akkermansia muciniphila* and *Faecalibaculum roden-
tium* (Supplementary Fig. 7e–h). Correlation analysis between micro-
bial abundance and environmental factors (HAI, DAI, colon length, and
body weight) based on the permutation test showed a significant
relationship between the gut microbiota phyla and the HAI, DAI, colon
length, and weight in colitis mice (Supplementary Fig. 7i, overall per-
mutation test *p* = 0.0011). There was a positive correlation between the
*Bacteroidetes* and HAI/DAI while a negative correlation with colon
length and body weight, which meant the increase of the *Bacteroidetes*
abundance was positively correlated with the exacerbation of colitis
after gavage of Pg. The *Firmicutes* were negatively correlated with HAI
and DAI but positively correlated with colon length and body weight,
consistent with the abundance of *Firmicutes* decreased following the
gastric administration of Pg exacerbating colitis. Besides, the angle
between the *Bacteroidetes* and HAI was significantly smaller than the
angle between the *Firmicutes* and the extension line of HAI, indicating
that the *Bacteroidetes* had a stronger correlation with environmental
factors than the *Firmicutes*. The correlation heat map also reflected
that the phylum *Bacteroidetes* had the strongest correlation with
environmental factors (Supplementary Fig. 7j). Fluorescence in situ
hybridization (FISH) showed that the level of *Bacteroidetes* bacteria
was higher in both the DSS+LiPg group and DSS+DePg group than in
the control group (Fig. 3g). Notably, the level of Pg in the colon also
increased, regardless of whether fresh Pg or its ultrasonic extract was
perfused (Fig. 3g and Supplementary Fig. 7c). The data above indicated
that the colitis in the DSS+LiPg group and DSS+DePg group was more
severe than that in the DSS + PBS group, which was probably due to the

more unbalanced gut microbiota composition characterized by an
increase in *Bacteroidetes* abundance (dominant) and a decrease in
*Firmicutes* abundance.

## Linoleic acid (LA) is the critical gut metabolite of dietary Pg in colitis mice that promotes Th17/Treg cell imbalance

Changes in microbial diversity and intestinal metabolites, including
fatty acids, amino acids and derivatives, play important roles in the
pathogenesis of IBD[23]. Some fatty acids are present in higher con-
centrations in the organs and plasma of specific-pathogen-free (SPF)
mice than in germ-free (GF) mice, which suggests that the gut micro-
biota is the origin of these unique fatty acids[24]. We observed that the
differences in the degree of colonic inflammation among the DSS +
PBS group, DSS+LiPg group and DSS+DePg group were mainly related
to the increase in the abundance of the phylum *Bacteroidetes* and the
decrease in the abundance of the phylum *Firmicutes*, rather than the
change in the levels of a certain species. Studies have shown that all
members of the *Bacteroides* phylum express AfAas2, which is essential
for the metabolism of long-chain fatty acids[25]. Gut lactic acid bacteria
can generate various fatty acids, most represented by the genus *Lac-
tobacillus* belonging to the phylum *Firmicutes*[24]. Thus, we proposed the
hypothesis that the Th17/Treg cell imbalance in the colitis micro-
environment may be closely related to the metabolic regulation of the
gut microbiota. To confirm our hypothesis, we employed untargeted
metabolomics analysis based on liquid chromatography with tandem
mass spectrometry (LC–MS/MS) to examine the faeces of DSS + PBS,
DSS + LiPg, and DSS + DePg mice. Orthogonal partial least squares
discriminant analysis (OPLSDA) showed that the point cloud plots
from these 3 group samples were distributed in different regions,
indicating that significantly different metabolites existed among the
groups (Supplementary Fig. 8a). Metabolic pathway enrichment ana-
lysis (Fig. 4a) revealed that the LA metabolism pathway differed most
significantly among these 3 groups. Furthermore, topological analysis
performed to calculate the extent of the impact of the metabolites of
interest on metabolic pathways (Supplementary Fig. 8b) showed that
the LA pathway was in the blue area (*p* < 0.05), indicating that the LA
metabolism pathway indeed played a key role in gut microbial meta-
bolic processes within these 3 groups. Heatmap cluster plots (Sup-
plementary Fig. 8c) showed significant inhibition of LA in the DSS+LiPg
and DSS+DePg groups compared to the control group. Histogram plot
analysis of up- and downstream metabolites also indicated that gavage
of Pg resulted in obvious inhibition of LA in colitis mice (Supplemen-
tary Fig. 8d–g).

To verify that LA was the key metabolite involved in the aggra-
vation of colitis as well as the imbalance of Th17/Treg cell after gavage
of Pg, we administered LA intragastrically into colitis mice (Fig. 4b). As
expected, LA significantly alleviated colitis based on weight loss, DAI,
colon length and HAI (Fig. 4c–h). Moreover, even if the gut microbiota
was cleared from colitis mice prior to administration of LA, LA could
still alleviate colitis (Supplementary Fig. 9a–g). That is, the gavage of Pg
caused a decrease in the levels of the gut microbiota metabolite LA and
then aggravated colitis, illustrating that LA was an end product of gut
microbiota metabolism. Consistent with this, LA treatment simulta-
neously decreased the percentage of Th17 cells, increased the

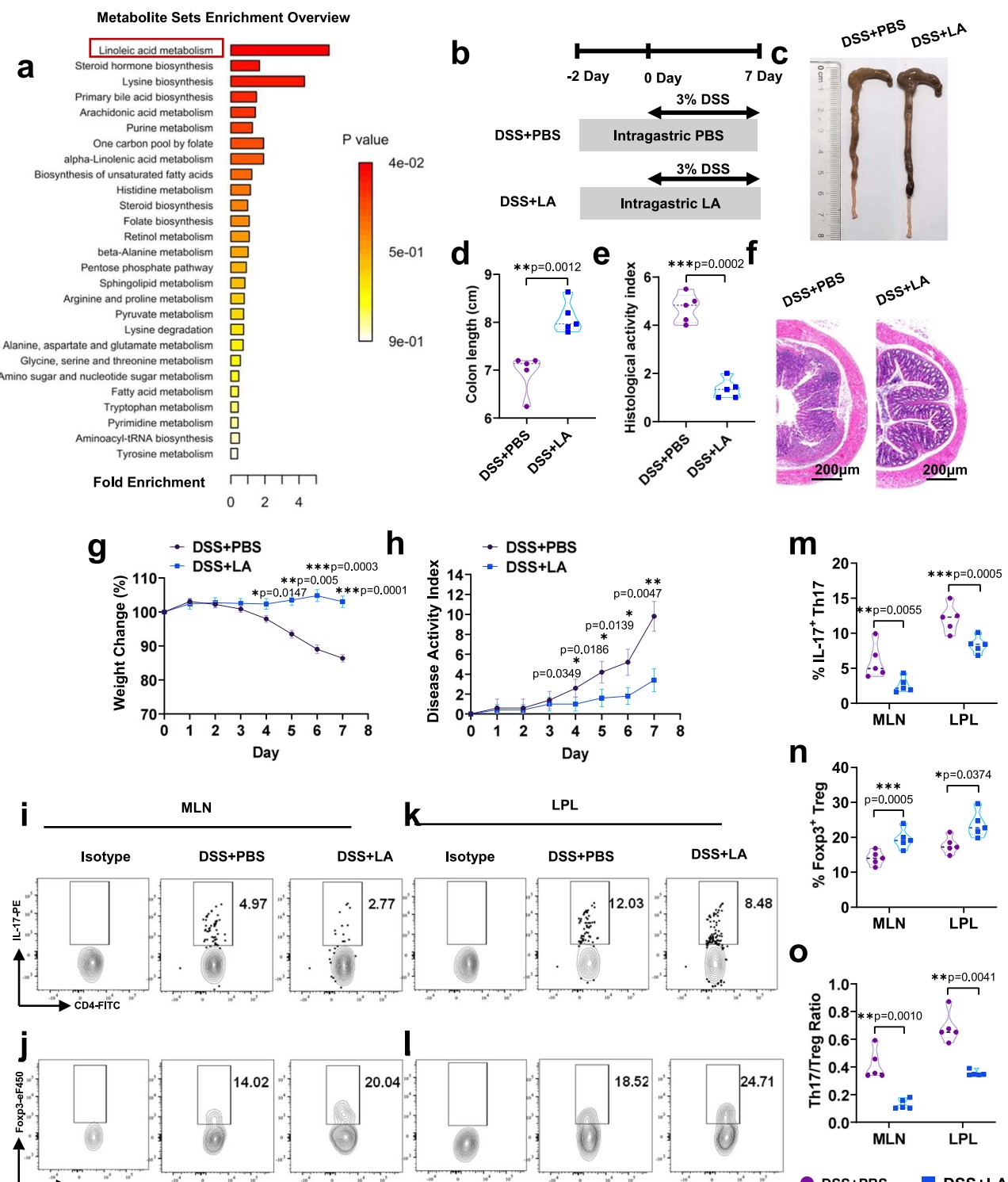

**Fig. 4 | Linoleic acid (LA) is the critical gut metabolite of dietary Pg in colitis mice and decreases the Th17/Treg cell balance. a** Enrichment analysis of the gut microbiota metabolic pathway from DSS + PBS, DSS + LiPg, and DSS + DePg mice based on untargeted LC–MS/MS. *n* = 3 biologically independent samples. **b** The experimental design. A DSS-induced colitis model was established. Colitis mice were administered PBS (DSS + PBS) or linoleic acid (DSS + LA). **c**–**f** Colon length (**c**, **d**) and histological score (**e**, **f**) of DSS + PBS and DSS + LA mice on day 8 after DSS induction, *n* = 5. Scale bar, 200 μm. **g** Body weight and (**h**) disease activity index

(DAI) of DSS + PBS and DSS + LA mice during DSS-induced colitis, *n* = 5 biologically independent samples. **i**–**l** The proportions of CD4+IL-17+ Th17 cells and CD4+Foxp3+ Treg cells and their ratio among total CD4+ T cells within the MLNs and LPLs were detected by flow cytometry and statistically analysed (**m**–**o**), *n* = 5. Data are presented as the mean ± SEM; *\*p* < 0.05; \*\**p* < 0.01; \*\*\**p* < 0.001 by two-sided unpaired *t* test. MLN mesenteric lymph node cells, LPL colon lamina propria lymphocytes, DSS dextran sodium sulfate. Source data are provided as a Source Data file.

percentage of Treg cells and corrected the Th17/Treg cell imbalance in MLNs and LPLs (Fig. 4i–o and Supplementary Fig. 9h–n).

The above results indicated that LA was the end metabolic product of Pg gavage that aggravated colitis and the Th17/Treg cell imbalance. Pg, as the key periodontitis pathogen, aggravates the colonic immune imbalance by interfering with gut microbiota composition and its metabolism.

## LA inhibited Th17 and promoted Treg cell differentiation in vitro

To explore the mechanism of the effects of LA on Th17 and Treg cell differentiation, we extracted mouse naïve CD4$^+$ T cells in vitro and measured the expression of IL-17 and Foxp3 under different induction conditions. After adding different concentrations of LA (0, 25, 50, 100, 200 μM) to medium with anti-CD3 and anti-CD28 beads for 3 days, the percentages of IL-17$^+$ Th17 cells and Foxp3$^+$ Treg cells among CD4$^+$ T cells were calculated by flow cytometry, and we found that neither IL-17 nor Foxp3 expression significantly differed among these groups (Fig. 5a, d, e, $p > 0.05$). When we additionally added 5 ng/ml TGF-β to induce Treg cell differentiation, the percentage of Foxp3$^+$ Treg cells in CD4$^+$ T cells showed a decreasing trend with increasing concentrations of LA (Fig. 5b, g, 25 μM, $p = 0.039$; 50 μM, $p = 0.001$) but did not cause differentiation of Th17 cells (Fig. 5b, f, $p > 0.05$). Considering that multiple cytokine pools exist within the colitis microenvironment and that it is a Th17-dominated inflammatory environment[26], we focused on the role of LA under Th17-inducing conditions in vitro. Thus, we added 2 ng/ml TGF-β and 30 ng/ml IL-6 to induce Th17 differentiation and simultaneously added different concentrations of LA (Fig. 5c). We found that when the concentration of LA reached 50 μM and above, the expression of IL-17 was inhibited, while Foxp3 expression was promoted in total CD4$^+$ T cells (Fig. 5h, i), indicating that under culture conditions favouring Th17 differentiation, LA could impair Th17 differentiation and promote conversion to Treg cells. These results were consistent with the in vivo finding that LA alleviated colitis by decreasing the Th17/Treg cell ratio. In parallel, we examined the proliferation and apoptosis of CD4$^+$ T cells in response to different induction conditions with LA coculture and found that different concentrations of LA did not affect their proliferation or apoptosis (Supplementary Figs. 6, 7), ruling out a possible influence of the overall number and activity of CD4$^+$ T cells on Th17 and Treg cell differentiation.

## LA regulated Stat1 activation via AHR to repress the differentiation of Th17 cells

To search for possible targets of LA in the regulation of Th17/Treg cell balance, we traced the results of differential KO (KEGG orthologous groups) using LEfSe metagenomic analysis of faeces from DSS + PBS, DSS+LiPg, and DSS+DePg group mice and showed that HSP90A (heat shock protein 90 alpha) and the AHR (aryl hydrocarbon receptor) complex were the characteristic gene biomarkers of the DSS+LiPg group in the Th17 differentiation signalling pathway (Supplementary Fig. 12a), suggesting that the decrease in LA as well as the increase in the Th17/Treg cell ratio in colitis mice mediated by gavage of Pg may be associated with AHR. Considering that AHR can regulate the differentiation of both Th17 and Treg cells depending upon the AHR ligand[27], we hypothesized that LA may be a ligand for AHR. To test this hypothesis, we used localized surface plasmon resonance (LSPR) to examine the binding between LA and AHR. The affinity constant of AHR and LA is 6.64e-5 M, indicating strong binding (Supplementary Fig. 12b). Furthermore, we detected the changes of AHR and HSP90 in the cytoplasm and nucleus after adding LA under the Th17-inducing conditions to confirm the dissociation of HSP90 and the nuclear translocation of AHR. Here, HSP90 was only detected in the cytoplasm, and its expression was increased in the con+LA group; the expression of AHR in the con+LA group decreased in the cytoplasm but increased in the nucleus (Supplementary Fig. 13a). These results indicated that

after LA bound to AHR, the HSP90 and AHR complex dissociated and then AHR entered the nucleus. At the same time, the expressions of AHR and AHR nuclear translocator (ARNT) under the non-reduced denaturing conditions were higher in the con+LA group, confirming that AHR bound to ARNT after entering the nucleus (Supplementary Fig. 13b).

To further examine the mechanisms regulating Th17 and Treg cell differentiation following LA and AHR binding, we used RT–qPCR to measure *Stat1*, *Stat3*, and *Stat5* expression in the con+LA group under Th17-polarizing conditions and found that LA promoted the expression of *Stat1* and *Stat5* and inhibited the expression of *Stat3* (Fig. 6a–c). Next, we extracted naïve CD4$^+$ T cells from the spleens of WT and AHR$^{-/-}$ mice, cultured them in vitro under Th17-polarizing conditions, measured Stat1/3/5 phosphorylation and total Stat1/3/5 protein by WB and found that only the phosphorylation of Stat1 at the Ser727 site was the downstream signalling event that followed the binding of LA and AHR (Fig. 6d–i). When we used RT–qPCR to measure the expression of *Ahr* at different time points after adding different concentrations of LA under Th17-inducing conditions, we found that the expression of *Ahr* began to be significantly inhibited from the 2nd day after adding LA (Supplementary Fig. 13c). Different concentrations of LA inhibited the expression of AHR (from 25 μM to 200 μM), especially at a concentration of 50 μM (Supplementary Fig. 13d). Thus, we speculated that LA was an antagonist of AHR. To further verify the above results, we cultured naïve CD4$^+$ T cells from WT and AHR$^{-/-}$ mice under Th17-polarizing conditions in vitro and measured IL-17 and Foxp3 expression on the 3rd day as well as p-Stat1 (Ser727) at 24 h after stimulation (Fig. 6j, k). We found that LA inhibited IL-17$^+$ Th17 and promoted Foxp3$^+$ Treg cell differentiation in an AHR-dependent manner (Fig. 6l, m). Consistent with this, once bound to AHR, the LA-AHR complex then prompts the phosphorylation of Stat1 at Ser727.

In summary, we demonstrated that under Th17-polarizing culture conditions, LA, by specifically binding to AHR as an antagonist, drove Stat1 phosphorylation at Ser727, which in turn repressed IL-17 while enhancing Foxp3 expression.

## LA supplementation alleviated Pg-induced aggravation of colitis and Th17/Treg cell imbalance in an AHR-dependent manner

To confirm that intragastric administration of Pg modulates the Th17/Treg cell imbalance and then mediates colitis via the LA-AHR metabolic pathway, we administered Pg to WT and AHR KO colitis mice. WT or AHR$^{-/-}$ mice were gavaged with fresh Pg or Pg+LA every 2 days starting 7 days before the establishment of the colitis model and continuing until the end of DSS administration. Mice were sacrificed on day 8 (Fig. 7a). Compared with WT colitis mice given Pg (WT (DSS+Pg) group), simultaneous administration of LA (WT (DSS+Pg+LA) group) significantly alleviated the reduction in colon length, weight loss, DAI and HAI (Fig. 7b–g), validating the above experimental findings that gavaging colitis mice with Pg resulted in acute reduction in the levels of the metabolite LA, which in turn aggravated colitis. Consistent with this, the WT (DSS+Pg+LA) group showed decreased percentages of IL-17$^+$ Th17 cells, higher percentages of Foxp3$^+$ Treg cells, and significantly lower Th17/Treg cell ratios in MLNs and LPLs than the WT (DSS+Pg) group (Fig. 7h–m and Supplementary Fig. 14a–d), further demonstrating that LA supplementation restored the Th17/Treg cell imbalance induced by intragastric Pg gavage in colitis, which also coincided with in vitro experiments showing that LA inhibited the differentiation of Th17 cells and promoted Treg cell differentiation under Th17-polarizing culture conditions. To verify that LA must associate with AHR to regulate colitis and the Th17/Treg cell balance, we utilized AHR KO mice and established a colitis model by administering either Pg or Pg + LA (Fig. 7a). There were no significant differences in colon length, body weight change, DAI, and HAI between the KO (DSS + Pg) group and KO (DSS+Pg+LA) group (Fig. 7b–g, $p > 0.05$), showing that LA supplementation did not ameliorate the exacerbation of colitis

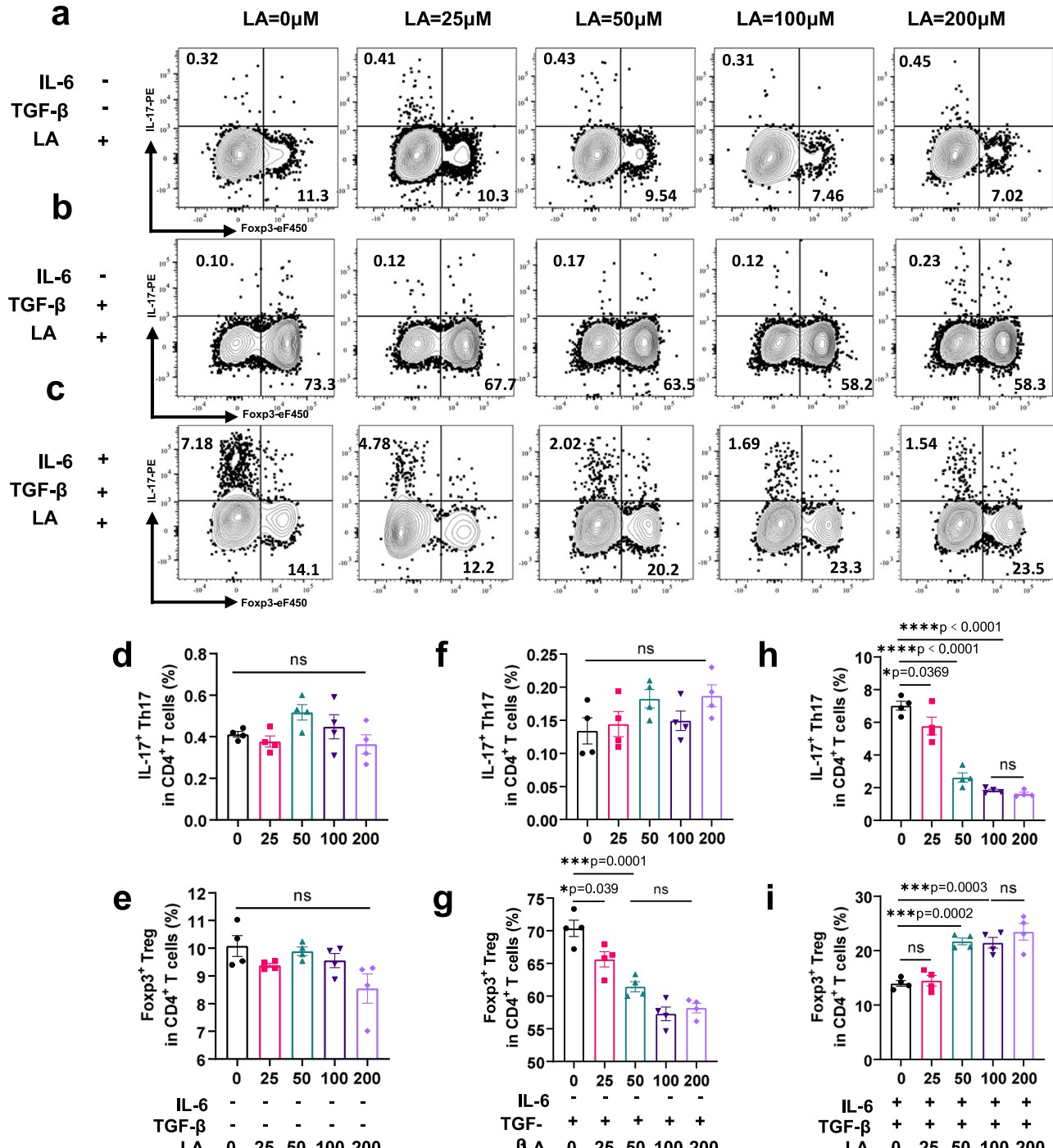

**Fig. 5 | Linoleic acid (LA) inhibited Th17 and promoted Treg cell differentiation in vitro. a** MACS-sorted naïve CD4[+] T cells were cultured with anti-CD3/CD28 beads for three days. The expression of IL-17 and Foxp3 in naïve CD4[+] T cells cocultured with different concentrations of LA detected by flow cytometry. **b** The expression of IL-17 and Foxp3 in naïve CD4[+] T cell culture conditions induced with TGF-β under different concentrations of LA was detected by flow cytometry. **c** The expression of IL-17 and Foxp3 in naïve CD4[+] T cell culture conditions induced with TGF-β and IL-6 under different concentrations of LA was detected by flow cytometry. **d**, **f**, **h** The proportion of CD4[+]IL-17[+] Th17 cells to total CD4[+] T cells in each group was statistically analysed. $n = 4$ biologically independent samples. **e**, **g**, **i** The proportion of CD4[+]Foxp3[+] Treg cells to total CD4[+] T cells in each group was statistically analysed. $n = 4$ biologically independent samples. Data are presented as the mean ± SEM; ns: no significant difference; *$p < 0.05$; **$p < 0.01$; ***$p < 0.001$ by two-tailed one-way ANOVA. Source data are provided as a Source Data file.

mediated by Pg under AHR-deficient conditions. Moreover, the lack of AHR in mice, which resulted in the inability of LA to bind to AHR, in turn blocked the signalling pathway that inhibited Th17 differentiation, as evidenced by the results that neither Th17 nor Treg cell percentages in MLN and LPL were significantly different between the KO (DSS+Pg) group and KO (DSS + Pg + LA) group (Fig. 7h–m and Supplementary Fig. 14a–d, $p > 0.05$).

In summary, we utilized AHR knockout mice to demonstrate that LA alleviated the colitis aggravation and Th17/Treg cell imbalance caused by Pg in an AHR-dependent manner in vivo. All of these findings

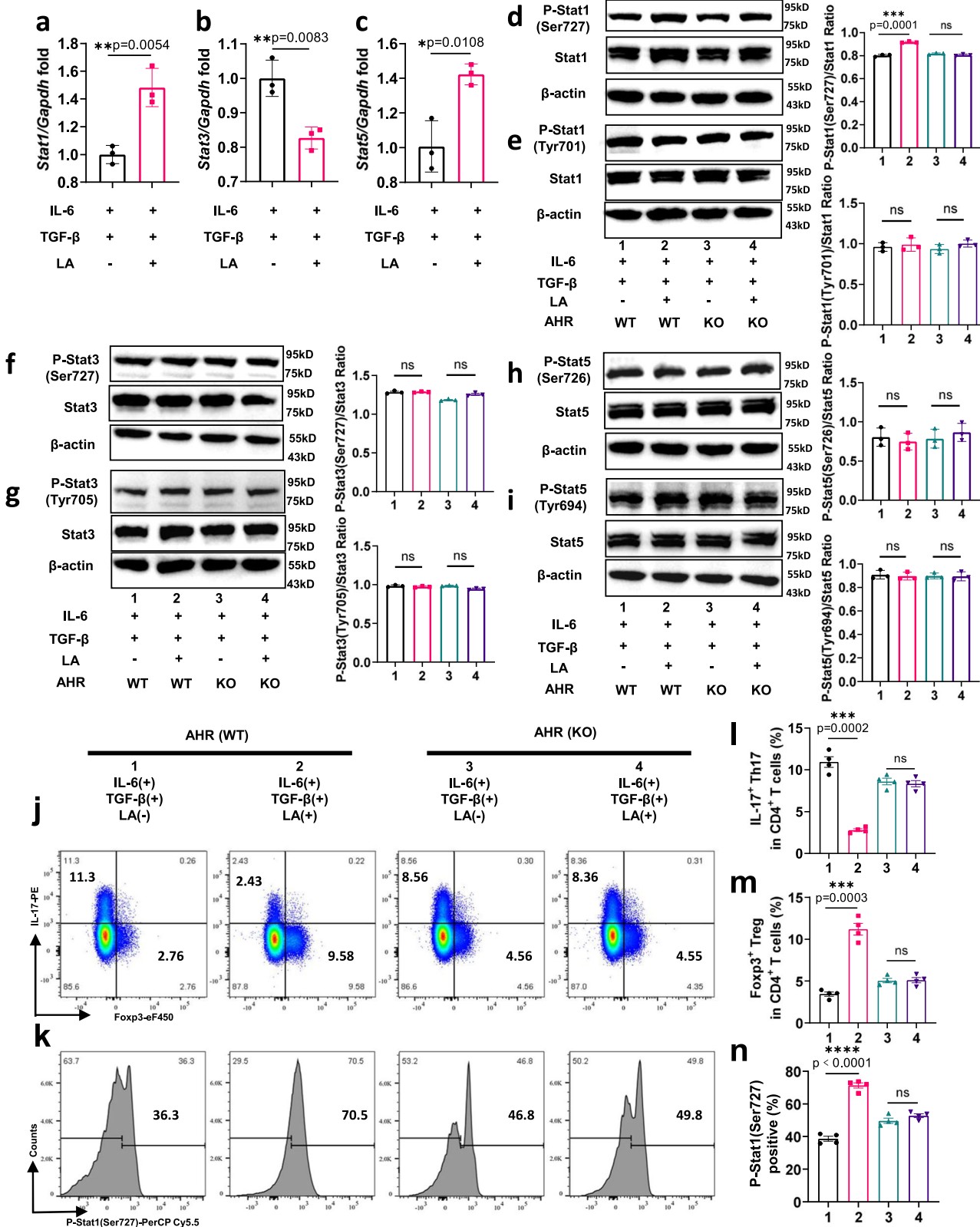

suggest that intragastric administration of Pg aggravates colitis via a gut microbiota-metabolite LA-Th17/Treg cell balance axis.

## Discussion

This study investigated the impact of crucial pathogenic bacteria in periodontitis, Pg, on colon inflammation in vivo. Our results indicated the significant damaging effect of Pg on DSS-induced colitis, as evidenced by aggravated weight loss and DAI score, short colon length and increased colon histology score. Pg exacerbated colitis in a gut microbiota-dependent manner through gut microbiota depletion and FMT. The gut microbiota exhibited increased microbial diversity and an increased proportion of the *Bacteroidetes* phylum as well as a decreased proportion of the *Firmicutes* phylum after Pg administration, which disfavoured the production of LA, thus improving the

**Fig. 6 | LA regulated Stat1 activation via AHR to repress the differentiation of Th17 cells. a–c** MACS-sorted naïve CD4⁺ T cells were cultured with anti-CD3/CD28 beads and stimulated with IL-6 plus TGF-β and 50 μM LA for three days. *Stat1, Stat3,* and *Stat5* gene expression was detected with RT–qPCR. *n* = 3 independent experiments. **d–i** Stat1, Stat3, Stat5, and proteins phosphorylated at different sites were detected by Western blotting after adding LA into the Th17-inducing cultures isolated from AHR WT and KO splenocytes. *n* = 3 independent experiments. **j** Naïve CD4⁺ T cells isolated from WT and AHR⁻/⁻ mice were stimulated with IL-6 plus TGF-β in the presence or absence of LA for three days, followed by restimulation with PMA and ionomycin cocktail for 6 h. Then, IL-17 and Foxp3 were detected by flow cytometry. **k** Naïve CD4⁺ T cells isolated from AHR WT and KO splenocytes were stimulated with IL-6 plus TGF-β in the presence or absence of LA for 24 h, fixed, permeabilized, and finally stained with phospho-Stat1 (Ser727). Intracellular levels of phospho-Stat1 (Ser727) were measured by flow cytometry. **l, m** The proportions of IL-17⁺ Th17 cells and Foxp3⁺ Treg cells to total CD4⁺ T cells in each group were statistically analysed. *n* = 4 biologically independent samples. **n** The positive rate of phospho-Stat1 (Ser727) was statistically analysed. *n* = 4 biologically independent samples. Data are presented as the mean ± SEM; ns no significant difference; *$p < 0.05$; **$p < 0.01$; ***$p < 0.001$ by two-tailed one-way ANOVA. MACS: magnetic-activated cell sorting. Source data are provided as a Source Data file.

Th17/Treg cell imbalance in the intestinal mucosa. Furthermore, we confirmed that LA regulated Stat1 activation via AHR to repress the Th17/Treg cell ratio. Our results support that intragastric administration of Pg aggravates colitis via a gut microbiota-metabolite LA-Th17/Treg cell balance axis.

Host genetics and the microenvironment influence the gut microbiota, a key player in mammalian physiology and pathology. Any changes in these factors can predispose the host to inflammatory disorders such as colitis[28]. Consistent with human IBD pathogenesis, the gut microbiota dysbiosis associated with the immune response is deeply involved in DSS-induced colitis[29]. Our results showed that gavage of Pg had differential effects on specific microbiota members (i.e., *Bacteroidetes* abundance increased while *Firmicutes, Verrucomicrobia,* and *Actinobacteria* abundance decreased) and thus likely changed the microbial metabolites and the host immune responses. The *Bacteroidetes* species has been studied as a biomarker of UC[30]. Previous studies revealed that oral administration of Pg induced an elevation of the population belonging to *Bacteroidetes* in the ileal microflora, which coincided with increased systemic inflammation[31]. Another study found that the administration of Pg significantly altered gut microbiota, with an increased proportion of phylum *Bacteroidetes* and a decreased proportion of phylum *Firmicutes*[32]. The *Firmicutes/Bacteroidetes* (F/B) ratio is widely accepted to have an essential influence in maintaining normal intestinal homeostasis since they are the two most critical bacterial phyla in the gastrointestinal tract. Decreased F/B ratio is regarded as dysbiosis, usually observed with IBD[33,34]. These studies were consistent with our data as we found that the aggravation of colitis caused by gavage of Pg exhibited characteristics of increased *Bacteroidetes* (dominated) and decreased *Firmicutes*. Usually, Pg was only transiently detected in the gut, where it failed to colonize[31]. The gut colonization by oral pathobionts and promotion of colitis required pre-existing intestinal inflammation (induced by DSS treatment) or a colitis-susceptible host[35]. In our experiments, when DSS was started, the actual detectable level of Pg in the intestine after gavage of Pg was already deficient (estimated to be 4 to 5 log10 units lower than the total gut bacterial counts). In fact, with shallow colonization levels (<0.01% of the total), Pg triggered changes to the commensal microbiota composition leading to inflammation. However, when homeostasis was compromised, Pg's role became minimal[36,37]. The above data suggest that the effect of Pg administration differs from other pathogenic bacteria, such as *Salmonella typhimurium*, that usually outgrow indigenous bacteria[32]. Further studies are needed to identify the molecules or metabolites derived from Pg that cause phenotypic and microbiological changes. In terms of the causal relationship of *Bacteroidetes* in exacerbating colitis, the expression levels of critical genes or molecules associated with regulatory functions related to excessive inflammation showed an increasing tendency alongside the increase in the proportions of vital *Bacteroidetes* species[38]. For example, *Bacteroides fragilis* induces Treg cell responses to ease colonic inflammation by altering the production of SCFAs[39], while our results showed that the increase in *Bacteroides caecimuris* abundance and the decreased levels of the metabolite LA in the colon were core features of the exacerbated colitis after gavage of live Pg. These gram-negative *Bacteroidetes* bacteria are thought to have lipid biosynthetic systems similar to those described in *Escherichia coli*, but little is known about their biology in general or their fatty acid metabolism[25]. Further studies of different *Bacteroidetes* species involved in fatty acid metabolism, especially LA, would be helpful to illuminate the detailed mechanism of the exacerbated colitis, which was also a limitation of this study. Therefore, we will perform in-depth follow-up studies.

One of the critical mechanisms by which the gut microbiota interacts with the host is through metabolites, which are small molecules produced as intermediate or end products of microbial metabolism[40]. Our studies found that gavage of Pg increased colonic Th17 and decreased Treg cell in a gut microbiota-dependent manner, and LA was the end metabolite under this specific condition. LA is an essential omega-6 polyunsaturated fatty acid (PUFA) and its presumed biological role was as a substrate for synthesizing arachidonic acid (ARA) via elongase and desaturase enzymes[41]. More evidence confirmed that LA was also a precursor to the bioactive lipid autacoids known as oxidized-LA metabolites (OXLAMs); however, its lipid-mediated roles in vivo remain understudied[42]. Here we found, LA, by specifically binding to AHR, drove Stat1 phosphorylation at Ser727, which in turn repressed IL-17 while enhanced Foxp3 expression. CD4⁺ T lymphocytes are crucial in the pathogenesis of IBD[43], and Th17 combined phenotypes of CD4⁺ T cells are found accumulated in the intestinal mucosa of IBD patients[44]. Besides, the imbalance between Th17 and Treg cells, which differentiate from CD4⁺ T cells, contributes to IBD[45]. Here, we confirmed that the increased Th17/Treg cell ratio was the cause of the exacerbation of colitis by gavage of Pg in a gut microbiota-dependent manner. Specific gut microbiota-derived metabolites have been implicated in producing proinflammatory cytokines and subsequent generation of Th17 cells[46,47]. Some commensal bacteria and their metabolites can also promote the generation of Treg cells, contributing to immune suppression[48]. To elucidate the exact mechanisms by which LA inhibited Th17 differentiation but promoted Treg cell differentiation, by combining metagenomic and metabolomic analyses, we found that the HSP90A/AHR complex was the characteristic gene biomarker in the Th17 differentiation signalling pathway following Pg gavage, prompting us to conclude that LA probably regulated Th17 differentiation through the AHR pathway.

AHR is widely expressed in immune and nonimmune cells of the gut and works as a potential target for controlling IBD[49]. The non-activated form of AHR is cytoplasmic, forming a complex with two HSP90s, a cochaperone p23, and XAP-molecule 2[50]. Dietary components and some chemicals or metabolites can activate AHR and induce the modulation of inflammatory responses[51]. Our results indicated that the decreased production of LA metabolites after Pg administration was responsible for aggravated colon inflammation. LA was a ligand for AHR with a strong binding force. Mechanistically, we confirmed that LA regulated Stat1 activation after binding with AHR to repress the differentiation of Th17 cells but promote the production of Treg cells. Consistent with previous findings, AHR was a negative regulator of IL-17-mediated signalling and inflammation in vitro[52]. Indeed, AHR exerts a different activity on Th17 or Treg cell differentiation dependent on

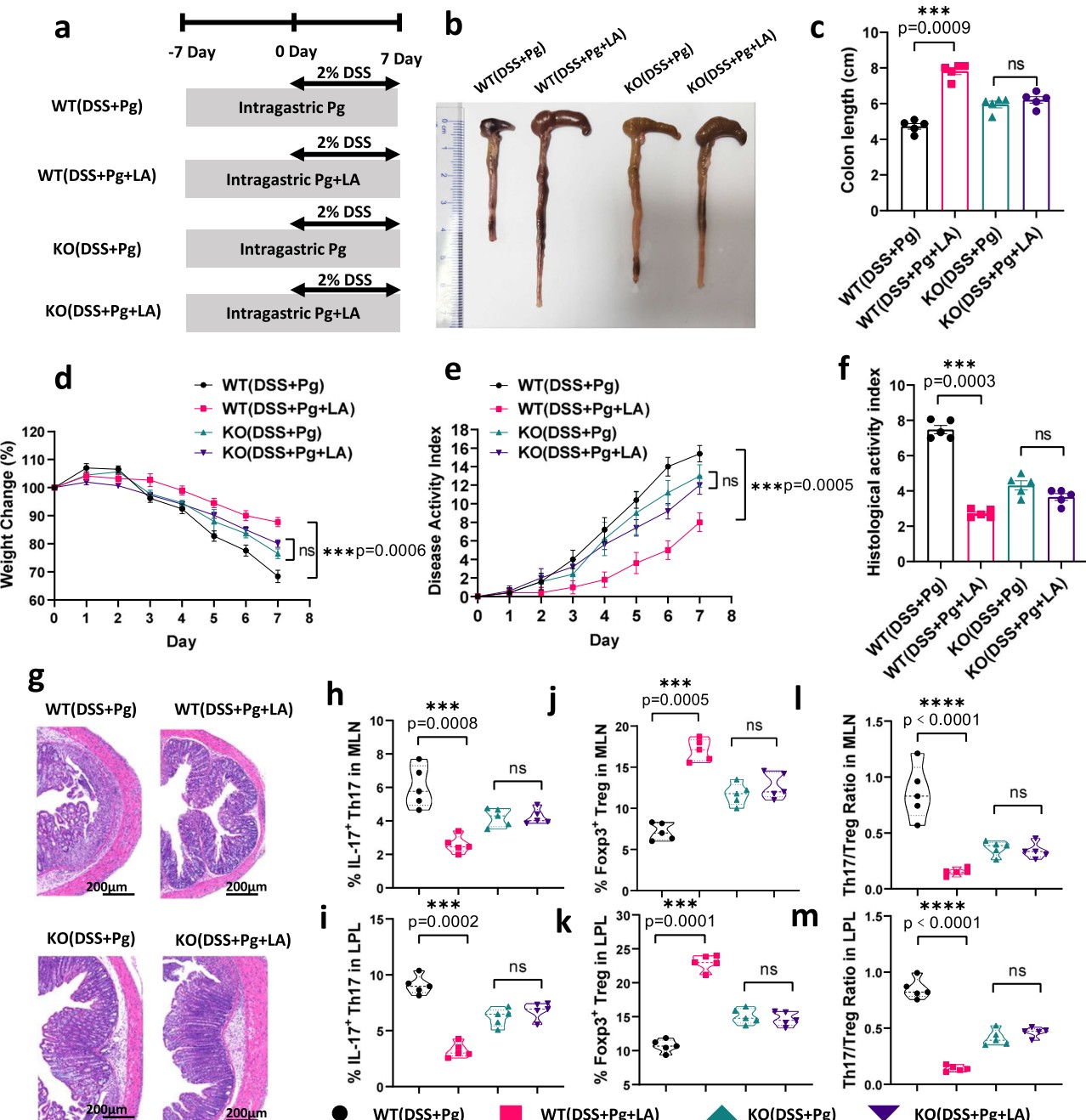

**Fig. 7 | LA supplementation alleviated Pg-induced aggravation of colitis and Th17/Treg cell imbalance in an AHR-dependent manner. a** The experimental design. Before the DSS-induced colitis model was established, WT or AHR KO mice were administered live Pg or Pg plus LA (named WT (DSS + Pg), WT (DSS + Pg + LA), KO (DSS + Pg), and KO (DSS + Pg + LA)). **b, c** Colon length of these four groups of mice on day 8 after DSS induction, $n = 5$. **d, e** Body weight (**d**) and disease activity index (DAI) (**e**) of mice during DSS-induced colitis, $n = 5$. **f, g** Histological scores of

the four group mice on day 8 after DSS induction. $n = 5$. Scale bar, 200 μm. **h–m** The proportions of IL-17+ Th17 cells and Foxp3+ Treg cells and their ratio among total CD4+ T cells within the MLNs and LPLs were detected by flow cytometry and statistically analysed. $n = 5$. Data are presented as the mean ± SEM; ns no significant difference; ***$p < 0.001$ by two-tailed one-way ANOVA. MLN mesenteric lymph node cells, LPL colon lamina propria lymphocytes, ANOVA analysis of variance, DSS dextran sodium sulfate. Source data are provided as a Source Data file.

different ligands[27]; for example, FICZ significantly promotes the differentiation and migration of Th17 cells, while L-Kyn has a significant effect on inducing the production of Treg cells[53]. The Stat family is essential for Th17 and Treg cell development. Moreover, Th17 cells are thought to be developmentally related to Treg cells[54,55]. Stat3 has a crucial role in Th17 differentiation because it is a critical positive regulator of RORγt and IL-17 expression[56,57]; Stat5 appears to act as a critical modulator of Treg cell development and function because IL-2/Stat5 signalling constrains Th17 production[58,59]. Previously, Stat1 was

reported to negatively regulate the differentiation of the inflammatory Th17 cell lineage[60]. Our experimental findings showed that following the binding of LA to AHR, activated Stat1 was phosphorylated at Ser727 to suppress IL-17 expression. However, we did not detect significant changes in Stat3 or Stat5 levels after IL-6 and TGF-β stimulation in AHR−/− mice. Consistent with previous results, AHR selectively interacted with Stat1 but not with Stat3 under Th17-polarizing conditions[61].

The deleterious effects of excessive AHR ligand degradation on intestinal immune functions could be counterbalanced by increasing

the intake of AHR ligands in the diet[62]. We have shown that the level of LA, as the AHR ligand, was significantly reduced in Pg-exacerbated colitis. Furthermore, we found that LA supplementation significantly ameliorated colon inflammation and Th17/Treg cell imbalance even after Pg gavage, indicating that LA was indeed the ultimate gut metabolite and was the critical targeting substance through which Pg interferes with colitis in an AHR-dependent manner. So, the altered faecal microbiota following Pg administration resulted in aggravated colon inflammation mediated through decreased metabolite LA production, which alleviated the Th17/Treg cell imbalance in an AHR-dependent manner during colitis progression.

In conclusion, our data demonstrated that critical pathogenic bacteria in periodontitis, Pg, aggravated colon inflammation in a gut microbiota-dependent manner. The underlying mechanism was associated with the improved Th17/Treg cell balance in the intestinal mucosa mediated by decreased microbiota-derived LA production. Our results revealed how Pg regulates colitis via a gut microbiota-LA metabolism-Th17/Treg cell balance axis. Our work provides a practical example of the interplay between the host and microbiota and gut metabolism during intestinal inflammation, highlighting the importance of LA-mediated immune regulation in preventing dysbiosis and intestinal inflammation.

## Methods
### Mice
$AHR^{-/-}$ mice were purchased from Model Organism (Shanghai, China) and were all generated from the same heterozygous $AHR^{+/-}$ parents with C57BL/6 background. AHR knockout mice were identified as homozygotes. C57BL/6 background mice (female or male) were purchased from SPF Biotechnology Co., Ltd. (Beijing, China). Littermates were randomly assigned to experimental groups. 8 to 10 weeks-old mice were maintained in pathogen-free conditions at the Laboratory Animal Centre of Beijing Stomatological Hospital, Capital Medical University (BSH-CMU) and provided with water and a standard laboratory diet *ad libitum* (12 h of light/dark cycle, 25 °C room temperature, 30 °C to 70 °C relative humidity). All animal experiments were performed in compliance with the relevant laws and approved by the Institutional Animal Care and Use Committee of Capital Medical University (No. KQYY-201712-002).

### Preparation of bacteria and its ultrasonicate
The Pg strain ATCC 33277 was revived on CDC anaerobic agar medium with 5 mg/l haematin chloride, 500 g/l vitamin K, 5% defibrillated sheep blood, and 0.1% L-cysteine hydrochloride and cultured under anaerobic conditions at 37 °C. The purity of the bacteria was verified by Gram staining and phase-contrast microscopy. Logarithmic growth phase Pg bacteria were collected and centrifuged at 14,000 rpm for 5 min at 4 °C. The bacterial cell pellet was washed twice with PBS and resuspended in a specific volume of sterile deionized water. The bacteria were thoroughly lysed using a JYD-150 ultrasonic cell disruptor (Zhisun Equipment Co. Ltd, Shanghai, China) at maximum power. The insoluble debris was removed by centrifugation, and the supernatant was sterilized by filtration through 0.22 μm-pore-size filters and stored at −80 °C.

### Oral administration of Pg or LA
A total of $10^9$ colony-forming units (CFUs) of live Pg suspended in 100 μl PBS or an equal volume of 50 μg/ml Pg extract was given to the oral cavity of each mouse through a feeding needle every other day for two weeks. The control group was sham administered 100 μl PBS without Pg. The number of bacteria administered was determined by considering the body weight and the number of bacteria in the saliva of periodontitis patients. For studies determining the effect of in vivo treatment of LA, mice (~50 days of age) were fed by gavage every other day for 9-14 days with 100 mg/day LA[63].

### FMT
8- to 10-week-old female C57BL/6 mice received antibiotic cocktails (ABX: vancomycin, 100 mg/kg; neomycin sulfate 200 mg/kg; metronidazole 200 mg/kg; and ampicillin 200 mg/kg) intragastrically once a day for 5 days for gut microbiota depletion. Faeces from donor mice (DSS + PBS, DSS + LiPg, DSS + DePg groups) were collected and resuspended in PBS at 0.125 g/mL, and then 0.15 mL of this suspension was administered to mice by oral gavage once a day for 5 continuous days.

### Experimental colitis and evaluation of DAI
Mice were administered DSS (3% w/vol; molecular weight: 36 to 50 kDa; MP Biomedicals) in drinking water for one week to generate a DSS-induced colitis model, followed by regular drinking water until the end of the study. When the antibiotic cocktails were administered for gut microbiota depletion, mice were administered 2% DSS to prevent preantibiotic-treated mice or KO mice from experiencing enhanced sensitivity[64]. Mice were sacrificed by $CO_2$ inhalation after DSS treatment, and colonic tissues were collected for further examination.

Clinical signs of colitis were evaluated based on the DAI scores, including body weight loss, occult blood, and stool consistency. Mice were scored blindly for the colitis experiments. In brief, the weight loss score was determined as follows: 0, no weight loss; 1, loss of 1–5% original weight; 2, loss of 6–10% original weight; 3, loss of 11–20% original weight; and 4, loss of >20% original weight. The bleeding score was determined as follows: 0, no blood by using Haemoccult (Beckman Coulter) analysis; 1, positive Haemoccult; 2, visible blood traces in stool; and 3, gross rectal bleeding. The stool score was graded as follows: 0, well-formed pellets; 1, semiformed stools that did not adhere to the anus; 2, pasty semiformed stool that attached to the anus; and 3, liquid stools that attached to the anus.

### Evaluation of the histological activity index (HAI)
Colons were fixed in 4% paraformaldehyde solution, embedded in paraffin, cut into 4 μm sections, and stained with H&E. The HAI was assessed according to the loss of goblet cells and crypts and the extent of inflammatory cell infiltration.

Crypt gland destruction was scored as follows:
0 points (normal)
1 point (a small amount of crypt loss)
2 points (a large amount of crypt loss)
3 points (extensive crypt loss)

Goblet cell loss was scored as 0 points (normal), 1 point (small), 2 points (large), and 3 points (most). Inflammatory cell infiltration was scored as 0 points (normal), 1 point (infiltration around crypt glands), 2 points (infiltration of the mucosal muscle layer), 3 points (extensive infiltration of the mucosal muscle layer with oedema), and 4 points (infiltration of the submucosal layer). The histological scores of each group were summed to obtain a total score of 0–10.

### Fluorescence in situ hybridization (FISH)
A segment of the mouse distal colon was fixed in 4% paraformaldehyde overnight at 4 °C, washed, and passed through 15% and 30% sucrose solutions. Colon tissues were then embedded in optimal cutting temperature compound (O.C.T., Tissue-Tek) and cryosectioned into 5 μm longitudinal sections (Leica). Slides were equilibrated in hybridization buffer (0.9 M NaCl, 20 mM Tris-HCl, 0.01% sodium dodecyl sulfate, 10% formamide, pH 7.5) and incubated in 10 ng/μl FISH probe (Takara) for 14 h at 42 °C in a humidified chamber. The probe information is listed in Supplementary Table 1. Slides were then incubated for 20 min in wash buffer (0.9 M NaCl, 20 mM Tris-HCl, pH 7.5) preheated to 42 °C and washed gently three times. Samples were then incubated in the dark with 10 μg/ml DAPI in PBS for 10 min at room temperature, washed three times with PBS, and mounted in an anti-fluorescence quenching blocking agent (Thermo). Images were acquired on a confocal microscope (KF-PRO-120, Beijing, China). Quantification was only

performed when the hybridization signals were strong and could clearly distinguish intact bacteria morphologically at a final observed magnification of 400× using the K-Viewer software. Fluorescence-labeled bacteria were counted in five randomly chosen microscopic fields per section, and the data for each group represented the mean of five quantifications.

### Isolation of naïve CD4⁺ T cells and T cell differentiation

Eight-week-old C57BL/6 mice were sacrificed by cervical dislocation, and the spleen was removed and ground with a 2 ml syringe needle in a precooled culture dish until a cell suspension was obtained. Then, the cell suspension was collected, and connective tissue was removed by filtering. This process was repeated twice until no mass was observed. Naïve CD4⁺ T cells were extracted following the instructions of the CD4⁺CD62L⁺ T cell Isolation Kit (Catalog no. 130-106-643, Miltenyi Biotec). The cell suspension was transferred into a 15 ml centrifuge tube, followed by centrifugation at 1600 rpm for 4 min at 4 °C. The precipitate was retained, followed by the addition of red cell lysate, and the mixture was then shaken and allowed to stand for 1 minute, followed by centrifugation at 1600 rpm for 4 min at 4 °C. Then, the residue was retained, and precooled PBS was added, followed by centrifugation again to collect the residue, which was finally prepared into a suspension by adding precooled PBS. Next, $1.0 \times 10^8$ cells were transferred into a 15 ml BD tube and centrifuged at 300 ×g for 5 min at 4 °C. The residue was retained and suspended in 0.5 ml buffer (PBS + 3% foetal calf serum), followed by the addition of 100 μl of biotin–antibody cocktail. The mixture was then blended and incubated for 45 minutes at 4 °C. Finally, 300 μl of buffer and 200 μl antibiotin MicroBeads were added and mixed, and the solution was allowed to stand at 4 °C for 30 minutes. Afterwards, the mixture was centrifuged at 300 ×g for 5 min at 4 °C, and the precipitate was collected and resuspended in 500 μl of buffer. The suspension was placed into the LS column (Miltenyi Biotec), and the filtrate was collected. The column was washed repeatedly to collect the filtrate. The filtrate was diluted, suspended, and finally stored on ice. Naïve CD4⁺ T cells were expanded in vitro and cultured using Dynabeads Mouse T-Activator CD3/CD28 (Invitrogen) for three days. As indicated, cultures were supplemented with recombinant cytokines: mouse IL-6 (30 ng/ml; Peprotech) combined with human TGF-β1 (2 ng/ml; Peprotech) for Th17 polarization and human TGF-β1 (5 ng/ml; Peprotech) alone for Treg cell polarization.

### Detection of proliferation and apoptosis

LA was added to the activated CD4⁺ T cell or induced Th17 or Treg cell culture system at final concentrations of 0, 25, 50, 100, and 200 μM.

For the proliferation test, parental CD4⁺ T cells were labelled with carboxyfluorescein succinimidyl ester (CFSE) by a CellTrace CFSE Cell Proliferation Kit (Catalog no. C34554, Invitrogen) according to the manufacturer's instructions; then, different concentrations of LA were added. Three days later, the CD4⁺ T cells were collected and detected by flow cytometry. The obtained data were analysed by ModFit LT software, and the proliferation index was statistically analysed.

For the apoptosis test, an Annexin V-PI Apoptosis Kit (Catalog no. 556547, BD Pharmingen) and flow cytometry were used to detect the apoptosis rate of CD4⁺ T cells in each group according to the manufacturer's protocols. The apoptosis rates were analysed by FlowJo software.

### Flow cytometric analysis of phospho-Stat1 (Ser727)

Naïve T cells were cultured with TGF-β plus IL-6 with or without LA for 24 h. The cells were fixed immediately by adding an equal volume of prewarmed Fixation Buffer from the True-Phos Perm Kit (Catalog no. 425401, BioLegend). The cells were incubated at 37 °C for 15 min to ensure that they were properly fixed. The cells were centrifuged at 350 ×g at room temperature for 5 min, and the supernatant was decanted. Sufficient Cell Staining Buffer (BioLegend) was added to wash the cells. While vortexing, cells were permeabilized by adding prechilled True-Phos Perm Buffer (BioLegend). The solution was incubated at −20 °C for at least 60 min to ensure the cells were properly permeabilized. The cells were centrifuged at 1000 ×g at room temperature for 5 min, and the supernatant was decanted and vortexed to resuspend the cell pellet. Cells were washed twice in Cell Staining Buffer and stained with PerCP/Cyanine5.5-conjugated phospho-Stat1(Ser727) antibody (1:100, Biolegend) for 1 h at room temperature. Flow cytometric analysis was performed with an Attune NxT 3 (Thermo).

### Measurement of p-Stat family expression by Western blotting

Cell cultures in each group were first treated with 1 mL containing protease inhibitor and phenyl-methyl-sulfonyl fluoride. Then, the lysate was centrifuged at 12,000 rpm for 10 min (L-500, Xiangyi Centrifuge Co., Ltd, Changsha, China). The supernatant was retained, and the protein concentration was measured using the bicinchoninic acid method. Next, the protein extract was resolved by 10% sodium dodecyl sulfate–polyacrylamide gel electrophoresis and blotted onto a polyvinylidene fluoride film. After blocking at 37 °C with 5% milk in PBS containing 0.1% Tween-20 for 2 h, the film was incubated with anti-p-Stat1(Ser727) antibodies (1:1000, Abcam), anti-p-Stat1(Tyr701) (1:1000, CST) antibodies, anti-p-Stat3(Ser727) antibodies (1:1000, CST), anti-p-Stat3(Tyr705) antibodies (1:1000, CST), anti-p-Stat5 (Ser726) antibodies (1:1000, Abcam), anti-p-Stat5(Tyr694) antibodies (1:1000, CST) and anti-β-actin antibodies (1:2000, Abcam) overnight. After extensive washing, the film was incubated with appropriate secondary antibodies (1:10,000, Jackson ImmunoResearch Laboratories) for 2 h, followed by development using the electrogenerated chemiluminescence method (Millipore, Billerica, MA). ImageJ software was used to calculate and statistically analyse the greyscale values of the WB bands. The monoclonal antibody clone names used in WB are shown in Supplementary Table 3 in the Supplemental information.

### Metagenome sequencing and analysis

Metagenomics is a means of directly interrogating the complete genomic information contained within a microbial population. Metagenomics bypasses the discipline of applying genomics techniques to study microbial communities in their natural environment by segregating and culturing microbial individuals. According to the manufacturer's instructions, total faecal bacterial DNA was extracted using a cetyltrimethylammonium bromide kit (Catalog no. 69525, QIAGEN). The DNA concentration and integrity were analyzed to screen qualified samples for sequencing. Individual libraries were constructed using the NEBNext Ultra DNA Library Prep Kit (Catalog no. E7103, New England Biolabs), and DNA sequencing was performed on the Illumina NovaSeq 6000 platform using a 2 × 150 bp paired end read protocol. Raw data were filtered with Trimmomatic to obtain high-quality clean reads[65]. The host DNA sequence was removed by alignment to the Mus musculus genome GRCm38.p3 with Bowtie 2 to obtain metagenomic DNA sequences. The metagenomic DNA sequences were assigned taxonomic labels using the Kraken 2 program[66] Then, Bracken (Bayesian Reestimation of Abundance after Classification with Kraken) was used to estimate the abundance of each sample at different phylogenetic levels (phylum, class, order, family, genus, and specie). Wekemo Tech Group Co., Ltd. (Shenzhen, China) completed the quantitative metagenomics analysis.

### Bioinformatics analysis

Following the generation of the species abundance table, functional abundance table, and abundance clustering analysis, a Venn diagram was used to illustrate the core microbiome at the species level. Nonmetric multidimensional scaling (NMDS) and principal coordinate

analysis (PCoA) based on the Bray-Curtis distance were used to perform the diversity analysis. Permutational multivariate analysis of variance (PERMANOVA) was applied to quantify the proportion of variation in taxonomy. The Dunn test was performed to detect significant differences in the relative abundance of different groups, and differences were considered significant when $p < 0.05$. The biomarkers of different groups were defined by LEfSe (linear discriminant analysis effect size, or LDA effect size) analysis to identify potential bacteria among different groups. The threshold on the logarithmic LDA score was set to 2.0 and 4.0. LEfSe first uses the nonparametric Kruskal–Wallis rank sum to detect species with significant differences in abundance between different groups. Then, the Wilcoxon rank-sum test was used to test the consistency of differences between different subgroups of different species from the previous step. Finally, LDA is used to estimate the magnitude of the impact of each component (species) abundance on the different effects.

### Metabolomics analysis

Metabolomics is a qualitative and quantitative analysis of all low molecular weight metabolites simultaneously in a given physiological period for a given organism or cell. Faecal samples from 3 animals from each group were used. The nontarget metabolome analysis of the samples was investigated using liquid mass spectrometry (LC-MS/MS). Samples were analysed by a Dionex Ultimate 3000 UHPLC coupled to a Q-Exactive high-resolution mass spectrometer (Thermo Scientific, USA). Chromatographic separations were accomplished using a $2.1 \times 50$ mm ZORBAX 1.8 μm C18 column (Agilent, USA). The mobile phases used were phase (A) water with 0.1% formic acid and 5 mM ammonium format and phase (B) methanol with 0.1% formic acid. The total run time was 30 min. The first 4 min of the run consisted of a linear gradient from 10% to 60% of B phase, followed by a 20-min linear gradient from 60% to 98% of B after reaching 98%, a stable run with 98% of B was sustained for 3 min followed by 3 min of 10% of B solution to regenerate the column. Solvents were pumped at 450 μl/min with a column temperature of 55 °C, and the sample chamber was held at 7 °C. Mass spectrometric measurements were acquired in positive ionization mode through a heated electrospray ionization (HESI) source. The nontargeted metabolomics data were obtained with the MS detector in full-scan mode (Full-MS) with data-dependent acquisition (dd-MS2) for the top 10 most abundant ions per scan. Mass spectrometer settings for full-MS were as follows: In-source CID 0.0 eV, Microscan's = 1, resolution = 70,000, AGC target 1e6, maximum IT = 50 ms, scan range 67 to 1000 m/z, spectrum data = Profile. The detector settings for dd-MS2 were as follows: Microscan = 1, resolution = 17,500, AGC target 1e5, maximum IT = 100 ms, loop count = 10, isolation window 2.0 m/z, NCE 15, 35, 50, spectrum data = profile, underfill ratio = 1.5%, charge exclusion = unassigned, and dynamic exclusion = 6 s[67]. Wekemo Tech Group Co., Ltd. (Shenzhen, China) completed the quantitative metagenomics analysis.

### Localized surface plasmon resonance (LSPR)

Surface plasmon resonance (SPR) analysis was conducted with an Open SPR instrument (Nicoyalife, Canada). The NTA sensor chip was first installed on the OpenSPR instrument by the standard procedure. (A). The buffer was run at the maximum flow rate (150 μL/min) and the bubble was exhausted after reaching the signal baseline. The running buffer was PBS. (B). The flow rate of the buffer solution (PBS, pH = 7.4) was lowered to 20 μL/min, and then 1-ethyl-3-(3'-dimethyl aminopropyl) carbodiimide (EDC) and N-hydroxysuccinimide (NHS) (1/1, mol/mol) solution were loaded to activate the NTA sensor chips. (C). A total of 200 μL of AHR protein solution was loaded, and the solution was run for 4 min. The injection port was rinsed with buffer solution and emptied with air. (D). The column was filled with 200 μL blocking solution (20 μL/min, 4 min), the sample ring was washed with buffer solution, and the device was emptied with air. (E). The baseline was

observed for 5 min to ensure stability. (F). Next, the selected compound, LA, was diluted into a series of solutions with different concentrations, which were then injected into the chip at low to high concentrations. The sample flowed through the chip at a constant flow rate of 20 μL/min. The binding time of compound-LA and the AHR protein was 240 s, and they naturally dissociated for 180 s. (G). The kinetic parameters of the binding reactions were calculated and analysed using Trace Drawer software (Ridgeview Instruments AB, The Kingdom of Sweden).

### Cytokine microarray assay

Colon explant cultures were performed according to a previous description[68]. The longitudinally opened colon was washed in ice-cold PBS containing 2× penicillin–streptomycin. Since different areas within the colon showed differences in the severity of colitis, we used a 3 mm skin punch tool to generate three to four defined circular biopsy samples from the same distal colon area for specific analysis. A single biopsy sample was transferred to a well of a 48-well plate containing 0.5 mL of sterile cell culture medium and incubated for 15 h in a cell incubator at 37 °C. The supernatant was collected and centrifuged to remove debris, and the cytokines were evaluated in QAM-TH17-1-1 cells according to the manufacturer's instructions (RayBiotech, Inc., Guangzhou, China).

### Analysis of gene expression in the colon

Total RNA from the colons was extracted using TRI Reagent (Takara). cDNA was synthesized with Transcriptor Universal cDNA Master Mix (Roche). Primers and probes specific to real-time PCR were purchased from Sangon Biotech Corporation. Reactions were carried out in a final volume of 25 μl in a LightCycler 96 System (Roche) using TaqMan Gene Expression Assays (Life Technologies Corporation) containing 900 nM each of the forwards and reverse primers and a 250 nM probe. The reactions consisted of a 10-min incubation at 95 °C, followed by 45 cycles of a 2-step amplification procedure consisting of annealing/extension at 60 °C for 1 min and denaturation for 15 s at 95 °C. LightCycler 96 software (Roche) was used to analyse the standards and carry out quantification. The relative levels of each mRNA were normalized to the relative levels of glyceraldehyde-3-phosphate dehydrogenase (GAPDH) mRNA[32]. The primer sets were made by Sangon Biotech and sequences of oligonucleotides are listed in Supplementary Table 2.

### Enzyme-linked immunosorbent assay (ELISA)

Colon tissue was weighed and placed into 900 mL normal saline followed by ultrasonic trituration and centrifugation at 3000 rpm for 10 min to obtain colon tissue homogenate. The levels of cytokines, including IL-10 (Dakewe, China), IL-17A (Dakewe), IL-6 (Dakewe), TGF-β (Dakewe), and IL-22 (Dakewe), in the colon tissue homogenate were measured by ELISA according to the manufacturer's instructions[69]. Capture antibodies (1:200) were coated in the plate overnight at 4 °C. After blocking, samples were incubated for two h at room temperature, followed by incubation with detection antibodies (1:200) for one h. Subsequently, streptavidin conjugated with horseradish peroxidase was added, and the substrate was added after 30 min. Finally, the absorbance value was detected using a microplate reader. The concentrations of cytokines were obtained according to the standard curves.

### Flow cytometric analysis of IL-17A and Foxp3

For staining of the intracellular cytokine IL-17A, cultured CD4 + T cells were first stimulated with Cell Activation Cocktail (with Brefeldin A) (BioLegend) for six hours in 5% CO2 at 37 °C. Subsequently, the cells were stained with a Zombie Yellow Fixable Viability Kit (1:300, Catalog no. 423103, BioLegend) for 15 min at RT in the dark. Then, FITC-labelled anti-CD4 antibodies (1:200, eBioscience) and AF700-labelled

anti-CD45 antibodies (1:100, Biolegend) were added for staining for 30 min at 4 °C in the dark, and the cells were fixed and permeabilized with Fixation/Permeabilization working solution (Biolegend) and Permeabilization Buffer for 20 min at RT in the dark. Finally, the cells were stained with PE-labelled anti-IL-17A antibodies (1:100, eBioscience). For the staining of transcription factor Foxp3 expression, cells were first stained with the Zombie Yellow Fixable Viability Kit (1:300, Biolegend). FITC-labelled anti-CD4 antibodies (1:200, eBioscience) and AF700-labelled anti-CD45 antibodies (1:100, Biolegend) were added to stain for 30 min at 4 °C in the dark. Subsequently, the cells were fixed and permeabilized with a Foxp3/transcription staining buffer set (eBioscience) for 30 min at RT in the dark. Finally, the cells were stained with an eFluor 450-labelled anti-Foxp3 antibody (1:100, eBioscience). Flow cytometric analysis was performed with an Attune NxT 3 (Thermo). FlowJo software was used for data analysis, and the monoclonal antibody clone names are shown in Supplementary Table 4 in the Supplemental information. The gating strategies for the FACS analysis of the percentages of IL-17$^+$Th17 cells and Foxp3$^+$Treg cells were shown in Supplementary Fig. 15 in the Supplemental information.

## Statistical analysis

All analyses were performed with GraphPad Prism 8 (GraphPad Software). Data are presented as the mean with SEM or violin plot. The Shapiro–Wilk test was used to verify the normality of the data. The difference between variables was evaluated by one-way analysis of variance (ANOVA) with Tukey's multiple comparisons test if data were normally distributed; otherwise, Kruskal–Wallis and Mann–Whitney $U$ tests were used to evaluate differences. The significant separation of the microbiome composition was evaluated by an analysis of similarities test. The correlation between two bacterial abundances was evaluated by linear regression analysis. Results with a $P$ value < 0.05 were considered statistically significant; asterisks denote p value results in the figures (*$p < 0.05$; **$p < 0.01$; ***$p < 0.001$ and ****$p < 0.0001$).

## Reporting summary

Further information on research design is available in the Nature Portfolio Reporting Summary linked to this article.

## Data availability

The metagenomic sequencing data have been deposited in the BioProject database at NCBI under accession number PRJNA907074 (http://www.ncbi.nlm.nih.gov/bioproject/907074). The Mus musculus genome GRCm38.p3 data used in the study is accessible from https://www.gencodegenes.org/mouse/release_M3.html. The authors declare that all other data supporting the findings of this study are within the article and its Supplementary Information file or available from the corresponding author upon request. A Figshare archive is present with this paper https://doi.org/10.6084/m9.figshare.24037005. Source data are provided with this paper.

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

## Acknowledgements

National Nature Science Foundation of China (81991504 and 81974149 to Y.L, 82122015 to J.X, 82101009 to L.J). National Key R&D of Program of China (Grant NO. 2022YFC2504200 to Y.L). Beijing Stomatological Hospital, Capital Medical University Young Scientist Program (YSP20200904 to L.J). Innovation Research Team Project of Beijing Stomatological Hospital, Capital Medical University (CXTD202202 to Y.L). Beijing Municipal Administration of Hospitals Clinical Medicine Development of Special Funding Support (ZYLX202121 to Y.L). Beijing Municipal Administration of Hospitals' Ascent Plan (DFL20181501 to Y.L). Beijing Municipal Administration of Hospitals' Youth Program (QML20181501 to L.G).

## Author contributions

Conceptualization: LJ and YL Investigation: YJ, LW, JF, JD, ZL and LG Methodology: LJ, YJ, LW, JF, JX Funding acquisition: LJ, LG, JX, YL Writing – original draft: LJ Writing – review & editing: JX and YL All authors read and approved the manuscript for publication.

## Competing interests

All authors declare no competing interests.

## Additional information

**Supplementary information** The online version contains Supplementary Material available at https://doi.org/10.1038/s41467-024-45473-y.

