## [Peer Review File · Nature Communications]

Porphyromonas gingivalis aggravates colitis via a gut microbiota-linoleic acid metabolism-Th17/Treg balance axisEditorial Note: Parts of this Peer Review File have been redacted as indicated to maintain the confidentiality of unpublished data.

REVIEWER COMMENTS

Reviewer #1 (Remarks to the Author):

This is an interesting study which employs a mouse model of colitis using DSS administration to demonstrate that gavage with the periodontal bacterium *Porphyromonas gingivalis* leads to exacerbation of colitis, an change in the microbial community composition of the gut microbiota and an associated shift in the intestinal Th17/Treg balance towards a more inflammatory phenotype. Furthermore, the authors use metagenomic and metabolomic approaches to indicate a decrease in the synthesis of linoleic acid (LA) in the altered microbiota and suggest that this reduction in LA is responsible for the shift in Th17/Treg cells by in vitro experiments which demonstrate that LA acts as a ligand of the aryl hydrocarbon receptor on naïve CD4+T cells and through this represses Th17 T cell differentiation and promotes Treg cell differentiation. In support of this mechanism the authors demonstrate that administration of LA to Pg challenged colitis mice reverses the enhanced disease indices and T cell balance in Pg challenged mice.

Overall, this is a well developed, thorough study which provides a potentially elegant mechanistic explanation for the aggravation of colitis by this periodontal organism in the DSS mouse model.

I have the following observations:

1. The manuscripts provides good evidence that administration of Pg caused increased disease in this colitis model system. Others – eg Flak et al JCI Insight 2019;4(13):e125191 – have linked oral gavage with the same organism in a mouse model to decreased gut barrier function and a reduction of pro-resolvin mediators in the intestine, specifically RvD5n-3 DPA. How does this shift in the balance of inflammatory mediators affect the outcome/interpretation of the current investigation. As an aside, it is not appropriate to refer to the sonicated supernatants of Pg as “Dead Pg” as to most readers this implies the use of killed whole cells. The antigenic challenge presented by whole cells versus sonicates is likely to be completely different and any differences observed in this investigation

requires very careful consideration.

2. The microbiological analyses demonstrate a shift in the balance of the composition of the gut microbiota following administration of *P. gingivalis*. What is not clear is the effect on the total microbial load in the intestine. In the equivalent studies in the oral cavity, gavage of healthy mice with Pg leads to a significant increase in the overall microbial numbers in the mouth which is likely to be influential in the resultant pathology. This may seem a rather trivial point but if administration of Pg in the current study led to a significant increase in the total microbial load then the a reduction in the overall metagenomic potential of the community away from LA biosynthesis may not be so consequential as it would be compensated for by increased metabolic activity overall.

3. Clearance of the microbiota using antibiotics prevented the pathological changes caused by administration of Pg. Did the authors determine whether residual antibiotics in these mice may have had an impact on the live Pg numbers in the intestine?

4. The FMT experiments demonstrate that the microbiota after Pg challenge was different to non Pg challenged mice and was able to cause intestinal pathology in recipient mice. Do the authors consider that this was a consequence of transfer of Pg in the FMT or the consortium as a whole?

5. The results of the microbial analysis of the faeces of Pg challenged versus non challenged mice report a significant elevation in the Bacteroidetes phylum in challenged animals. How much of this increase was a consequence of the presence of Pg – itself a member of the Bacteroidetes?

6. Did the metabolomic assays determine the actual change in concentration of LA in the intestines of the differently treated animals. This is relevant to Point 2 above but also to the interpretation of the dose response in vitro experiments which indicate that 50uM was an important threshold.

Reviewer #2 (Remarks to the Author):

In this manuscript, Jia et al demonstrate the protective effect of Linoleic acid (LA) in colitis. *Porphyromonas gingivalis* (Pg) acted as a critical pathogen in colitis and Pg-induced gut dysbiosis led to decreased LA production. LA administration ameliorates colitis in DSS model through controlling Th17/Treg balance (decrease Th17, increase Treg population). In the

details, LA suppressed Th17 population through regulating STAT1 activation via AHR. This is a straightforward study that demonstrate protective effect of LA in colitis using in vitro and in vivo system. The conclusions are largely supported by the presented data and the experimental design is overall sound. However, there is an issue that need to be clarified before the manuscript can be considered any further:

Concerns:

1. In many facs polts, gating is not correct. For example, in Figure 1J, gating does not include all IL-17 positive population.
2. In Extended Data Fig 6, how to analysis cell proliferation? Which method did you use? Authors should mention that and explain in method section.
3. In Extended Data Fig 7, how to analysis apoptosis? Which method did you use? Authors should mention that and explain in method section. And X and Y axis labeling is missing.
4. If authors show gating strategy for facs analysis, it will be more informative. Especially, analysis for MLN and LPL
5. Authors used different concentration of DSS (2%, 3%). Is there any reason?
6. In Figure 6, the expression of STAT5 increased by AHR antagonist and no changed by LA treatment. What is the different MOA of LA/AHR between STAT1 and STAT5?

Reviewer #3 (Remarks to the Author):

In this paper, the authors have elucidated the causative relationship between periodontitis and colitis and demonstrated that the periodontal pathogen-*Porphyromonus gingivalis* (P.g.) aggravates colitis by increasing the abundance of Bacteroides that reduces the level of linoleic acid. Linoleic acid binding to AHR results in the increased phosphorylation of Stat1, leading to the decreased ratio of Th17/Treg. The authors presented a detailed molecular mechanism to clarify the association between periodontitis and colitis; however, additional data and clarification are needed to improve the manuscript and strengthen the conclusion. Furthermore, the manuscript has many mistakes; the authors should prepare it more carefully before submission.

Major issues:

Experimental design:

The authors have administrated P.g. by oral gavage to investigate the pathological relevance between periodontitis and IBD. This method allows them to analyze the effect of P.g. on colitis, but not suitable for determining the pathological relevance of periodontitis. To rigorously examine it, they should orally infect mice with P.g. to develop periodontitis and then investigate susceptibility to experimental colitis.

Introduction

L.75-76: The term “metabolic disorder” represents "illness" related to the metabolic system, such as diabetes and obesity. However, the authors deem this term to indicate the alteration in microbial metabolism caused by the altered gut microbial composition, which may mislead the readers.

Moreover, the metabolite alteration is poorly mentioned in the introduction. The authors should provide further evidence on the association between metabolite alteration and periodontitis/IBD.

Result

In Fig.1, It remains obscure whether P.g. colonizes and expands in the intestinal lumen. The authors should show the abundance of P.g. in the intestine after seven days of oral gavage of P.g.

In Fig. 1l-n, the authors have claimed that the exacerbation of colitis by gavage of P.g. depends on the increased ratio of Th17/Treg. However, the increased Th17/Treg ratio may be a consequence, but not a cause, of the exacerbation of colitis, given that they analyzed it after the development of colitis (Day 7). The authors should reanalyze Th17/Treg balance before the onset of colitis (e.g., Day 2).

Moreover, although it is controversial, DSS-induced colitis does not depend on Th17 cells (DOI: 10.4110/in.2011.11.6.416). Activation of macrophage/monocytes plays a vital role in the pathogenesis of this colitis model. Therefore, the authors should carefully analyze myeloid cell subsets, including monocytes, CX3CR1^{low} (inflammatory), and CX3CR1^{high} (anti-inflammatory) macrophages at the early stage of inflammation.

In Fig. 2, the authors have stated that the exacerbation of DSS-induced colitis and Th17/Treg

imbalance are attributed to the alteration of gut microbiota. Nevertheless, the FMT experiment does not exclude the possibility that P.g. directly influences the inflammatory status and T-cell balance, considering that donor feces is likely to contain P.g. (Fig. 3g).

In Fig2A, the authors should show gut microbial composition after the FMT to confirm the reconstruction of donor microbiota.

In Fig.3c, the gut microbial composition should be shown not only at the phylum level but also at the species level to provide the relative abundance of P.g..

In Fig3F, It is an overstatement that Bacteroides are responsible for the exacerbation of colitis. To conclude this, the authors need to perform experiments using gnotobiotic mice associated with AfAas2 knocked-out Bacteroides caecimuris.

In Fig. 3g, Because Pg is also detected in the DePg group, there are non-specific signals. The authors should reanalyze it using more specific probes or at least explain it.

In L. 148-149 (Fig. 3b), the authors have described that the samples from the DSS+LiPg group show lower similarity than those from the DSS+DePg group in species composition structure compared with those from the control group. To discuss the similarity of microbial structure among the groups, they should quantify the distance between the samples and show the re in a box plot.

L. 165-168, 174-175. the authors have stated that the severity of colitis may be due to the more unbalanced gut microbiota composition and the significant increase of Bacteroides. However, as pointed out above, the possibility that Pg itself may exacerbate colitis has not been excluded. It is not sure that the degree of colitis is related to the expansion of Bacteroides. Even so, it is uncertain why the authors focused on metabolites. The information "Studies have shown that all members of the Bacteroides phylum express AfAas2, which is essential for the metabolism of long-chain fatty acids" (lines 175-177) should be provided before the hypothesis (lines 173-175).

In Fig. 5, do the LA concentrations in the in vitro experiments reflect physiological amounts?

Please provide the rationale for how the authors determined the concentrations.

L. 248-252, this statement is not convincing because the protein expression levels are comparable between the absence or presence of LA (Fig. 6f, h, g, i).

Fig. 7, To strengthen the authors' conclusion, T-cell-specific Ahr KO mice such as Lck-Cre Ahrflox/flox mice (doi: 10.1073/pnas.1111786108; 10.1073/pnas.0504757102) should be employed. Considering that various cell types, including dendritic cells and intestinal epithelial cells, express Ahr (doi: 10.1016/j.immuni.2017.12.012, 10.3389/fimmu.2021.638725), germline KO of Ahr may exert considerable influence on these cell types to affect Th17/Treg balance indirectly.

Fig.8 indicates that the binding of LA to Ahr facilitates the dissociation of HSP90 and the nuclear translocation of Ahr. However, no experimental evidence supports this model. Also, it remains unclear how LA-bound Ahr increases phosphorylated STAT1.

Discussion

In L. 308-309, the authors have mentioned the decreased microbial diversity; however, it is difficult to conclude without data on the alpha diversity of gut microbiota.

In L. 324 and 464, the reference to explain the reason why they reduced the concentration of DSS is the secondary source. Please quote the original citation (doi: 10.1538/expanim.50.387).

In L. 315-324, it seems unnecessary to discuss the methodology, such as the concentration of DSS, because it is out of the focus of the current study. Please omit it or describe it briefly.

In L. 352-354, please cite appropriate references.

In L. 374-378, the data using an Ahr antagonist are not convincing. Quantification of the western blotting data is desirable. It is also possible to speculate that LA may serve as an antagonist for Ahr. Does an Ahr agonist cancel LA-mediated Th17/Treg imbalance?

The mechanism by which Pg increases Bacteroides is still unclear. The authors should at least discuss it.

Minor points (Grammar and Typo)

The manuscript contains numerous grammar errors. English editing by a native speaker is strongly recommended.

The format of the gene name should be in italic throughout the manuscript.

The abbreviations have double notations (IBD in L43 and L47; Pg in L74 and L84).

L.55: "RoRy" should be "RORy."

L.53: "T helper cell 17" should be "T helper 17 cell."

L85: "WT" should be spelled out at the first appearance.

L90: "Extended Data Fig.2a-d" may be "Extended Data Fig.2A-D" following the description in Figures.

Fig1,2 and others; The authors should display the values in a row.

L.212, 217-218; Please add a space before the units.

L156,236; "LEFSe" should be "LEfSe."

Extended Data Fig6A-C, 7A-C; The authors should enlarge the panels and the axis labels.

L.511,513 "300 g" should be "300×g."

L.516; 8 of "1.0×10⁸" should be superscript.

L.572-573; In metagenome analysis, why did you remove the host DNA sequence by aligning it to the human genome (hg38)? The authors should use Mus musculus assembly like mm10.

L.587; The authors should use multiplication signs, not an alphabet.

L221-227; Fig5C, H, I are not described.

Fig3G; The authors should explain how to quantify these bacteria in more detail.

Method

Please describe the method of LEfSe.

The authors should perform a variance test before examining statistical differences.

L.681-682; Did the author evaluate the correlation between two bacterial abundance? If not, the authors should omit this method.

Please describe the clone names of the monoclonal antibodies.

Point-by-point Response

Reviewer #1:

This is an interesting study which employs a mouse model of colitis using DSS administration to demonstrate that gavage with the periodontal bacterium *Porphyromonas gingivalis* leads to exacerbation of colitis, a change in the microbial community composition of the gut microbiota and an associated shift in the intestinal Th17/Treg balance towards a more inflammatory phenotype. Furthermore, the authors use metagenomic and metabolomic approaches to indicate a decrease in the synthesis of linoleic acid (LA) in the altered microbiota and suggest that this reduction in LA is responsible for the shift in Th17/Treg cells by in vitro experiments which demonstrate that LA acts as a ligand of the aryl hydrocarbon receptor on naïve CD4+T cells and through this represses Th17 T cell differentiation and promotes Treg cell differentiation. In support of this mechanism the authors demonstrate that administration of LA to Pg challenged colitis mice reverses the enhanced disease indices and T cell balance in Pg challenged mice.

Overall, this is a well-developed, thorough study which provides a potentially elegant mechanistic explanation for the aggravation of colitis by this periodontal organism in the DSS mouse model.

R: We appreciate the reviewer's comments and suggestions. We thoroughly considered all points raised by the reviewer and performed additional experiments to answer them.

1. The manuscript provides good evidence that administration of Pg caused increased disease in this colitis model system. Others – eg Flak et al JCI Insight 2019;4(13):e125191 – have linked oral gavage with the same organism in a mouse model to decreased gut barrier function and a reduction of pro-resolvin mediators in the intestine, specifically RvD5n-3 DPA. How does this shift in the balance of inflammatory mediators affect the outcome/interpretation of the current investigation.

R: We thank the reviewer for the thoughtful questions. Flak, M. B. et al. demonstrated that inoculation with *P. gingivalis* to inflammatory arthritis promoted gut barrier breakdown, downregulated intestinal RvD5_{n-3} DPA, and exacerbated joint inflammation[1]. As they proposed, gut homeostasis and barrier function is recognized as critical in inflammatory diseases[2]. Similarly, we found that administration of *P. gingivalis* to colitis altered the gut microbiota composition, downregulated intestinal LA, and exacerbated colitis inflammation. Both RvD5_{n-3} DPA and LA, the bioactive lipid autacoids, can exert anti-inflammatory effects through AHR, providing a favorable basis for exploring the biological regulation mechanism of lipid regulators. Based on the pioneering theory of converting the growing knowledge of the host-microbiota relationship into therapeutic approaches as proposed by Flak, M. B. and co-workers, we made additional discussion and cited these references to give readers a clearer understanding of the biological activity effects of lipid mediator-LA in Lines 431-437 (References 41-42).

2. As an aside, it is not appropriate to refer to the sonicated supernatants of Pg as “Dead Pg” as to most readers this implies the use of killed whole cells. The antigenic challenge presented by whole cells versus sonicates is likely to be completely different and any differences observed in this investigation requires very careful consideration.

R: We agree that referring to Pg ultrasonic extract as “dead Pg” may cause readers to misunderstand. We have deleted the word “dead” in the manuscript, including Fig. 1 and Supplementary Fig. 1.

3. The microbiological analyses demonstrate a shift in the balance of the composition of the gut microbiota following administration of *P. gingivalis*. What is not clear is the effect on the total microbial load in the intestine. In the equivalent studies in the oral cavity, gavage of healthy mice with Pg leads to a significant increase in the overall microbial numbers in the mouth which is likely to be influential in the resultant pathology. This may seem a rather trivial point but if administration of Pg in the current study led to a significant increase in the total microbial load then a reduction in the overall metagenomic potential of the community away from LA biosynthesis may not be so consequential as it would be compensated for by increased metabolic activity overall.

R: We thank the reviewer for this thoughtful question. In our experiment, administration of Pg to colitis mice did not cause statistically significant changes in the total gut microbiota load. Here, we showed the alpha diversity analysis of the fecal bacteria from the DSS+PBS, DSS+LiPg, and DSS+DePg group mice. The Chao1 index, which calculated the community richness based on the number of OTUs, showed no significant difference among the three groups, confirming that administration of Pg didn't affect the total microbial load in the intestine (Please see the Response-Fig. 1a-c, $p > 0.05$). Besides, the Simpson index calculating the community diversity showed a significant difference among the three groups (Response-Fig. 1d-f, $p=0.027$), further confirming that the DSS colitis after gavage of *P. gingivalis* was more severe probably due to the altered gut microbiota composition rather than the total number of OTUs in the community.

In the equivalent studies from Hajishengallis, G. et al., they found that introducing *P. gingivalis* into normal SPF mice caused elevation of the total cultivatable commensal bacterial load and changed the qualitative composition of this microbiota, leading to inflammatory periodontal bone loss[3]. They also found colonization of germ-free (GF) mice by *P. gingivalis* failed to induce bone loss. Their study emphasized the role of the entire oral microbiota in the cause of pathogenic bone loss, not only the numbers of the total commensal bacteria. This viewpoint shares similarities with our experiment.

In fact, the impact of administrating a certain bacterium on the total bacterial load in recipient mice has been controversial in different experiments, for example, some studies have shown that it may cause suppression of the commensals[4]. These differences may be related to the characteristics of the inoculation bacteria themselves and the state of the recipient mice.

[Figure redacted]

Response-Fig. 1. The alpha diversity analysis of the fecal bacteria from the DSS+PBS, DSS+LiPg, and DSS+DePg group mice. (a) The Chao1 boxplot to calculate the community richness based on the number of OTUs from the DSS+PBS, DSS+LiPg, and DSS+DePg group mice feces. (b-c) The p-value of all groups or pairwise of the Chao1 index after the Kruskal-Wallis's test. (d) The Simpson boxplot to calculate the community diversity from the DSS+PBS,

DSS+LiPg, and DSS+DePg group mice feces. (e-f) The p-value of all groups or pairwise of the Simpson index after the Kruskal-Wallis's test. $p < 0.05$, significant difference.

4. Clearance of the microbiota using antibiotics prevented the pathological changes caused by administration of Pg. Did the authors determine whether residual antibiotics in these mice may have had an impact on the live Pg numbers in the intestine?

R: We thank the reviewer for this thoughtful question. To determine whether residual antibiotics in these mice have an impact on the live Pg numbers in the intestine, we designed the following experiment (Response-Fig. 2a). The Pg content in the feces of mice in the DSS+Pg group reached its peak on the 3rd day (Response-Fig. 2b, Day 3, $2.14 \times 10^9 \pm 3.29 \times 10^8$) and began to decrease from the 5th day (Day 5, $1.27 \times 10^8 \pm 2.28 \times 10^7$). The Pg content in the feces of mice in the ABX(DSS+Pg) group continually remained a very low level (Response-Fig. 2b, Day 5, $3.59 \times 10^7 \pm 1.75 \times 10^7$; Day 7, $2.29 \times 10^8 \pm 6.14 \times 10^7$). The above data indicated that antibiotic treatment significantly reduced the amount of Pg in the intestine, especially in the first seven days. Besides, we can see that the content of Pg showed no statistical difference between the two groups from the 7th day to the 14th day ($p > 0.05$).

In our experiment, using antibiotics to clear the gut microbiota was able to eliminate the aggravation of colitis caused by the gavage of Pg extract, indicating that the essence of affecting the degree of inflammation was not related to the biological activity of Pg itself but to the entire gut microbiota. We added these data as revised Supplementary Fig. 4a-b and revised the manuscript correspondingly in Lines 153-158.

[Figure redacted]

Response-Fig. 2. Residual antibiotics reduced the content of Pg in mice feces. (a) Diagram of this experiment. WT mice were given by gavage of the quadruple antibiotic cocktails for gut microbiota depletion (ABX(DSS+Pg) group). A total of 1×10^9 CFUs of live Pg suspended in 100 μ L PBS was given to each mouse by gavage through a feeding needle every other day for 14 days. DSS was administered from the 7th day. (b) Line chart of Pg absolute copies per gram of mice feces in both groups by qPCR. $n=5$. Data are presented as the mean \pm SEM; ** $p < 0.01$; *** $p < 0.001$ by t-test.

5. The FMT experiments demonstrate that the microbiota after Pg challenge was different to non Pg challenged mice and was able to cause intestinal pathology in recipient mice. Do the authors consider that this was a consequence of transfer of Pg in the FMT or the consortium as a whole?

R: We thank the reviewer for this professional question. We believed that the aggravation of colitis after FMT was a consequence of the consortium, not the transfer of Pg only. To prove this, we designed the following experiments (Response-Fig. 3a). Firstly, the content of Pg in the feces of DSS+LiPg group mice was quantitatively detected by qPCR and labeled as Pg[#], which was about 2.65×10^8 /g as answered in Question 4 (please see Response-Fig. 2). The same Pg[#] bacterial load was added to normal mouse fecal suspension and labeled as Nor+Pg[#]. The gut microbiota-depleted recipient mice (GDR mice) were reconstituted with equal volumes of Pg[#] liquid (FMT(Pg[#]) group),

or Nor+Pg[#] (FMT(Nor+Pg[#]) group), or the fecal suspension from DSS+LiPg-treated mice (FMT (DSS+Pg) group) via intragastric administration. FMT(DSS+Pg) group mice developed more severe inflammation than FMT(Pg[#]) group and FMT(Nor+Pg[#]) group as judged by weight loss, DAI, colon length, HAI (Response-Fig. 3b-g), suggesting that the enhanced colitis in GDR mice were caused by changes in gut microbiota through consumption of feces from the DSS+LiPg group mice. Moreover, FMT(Pg[#]) group mice developed almost the same degree of inflammation compared with FMT(Nor+Pg[#]) ones (Response-Fig. 3b-g, $p > 0.05$), indicating that Pg in the FMT was not the dominant factor leading to intestinal pathology. We added these data including Response-Fig. 2 and 3 as revised Supplementary Fig. 4 and revised the manuscript correspondingly in Lines 153-164.

[Figure redacted]

Response-Fig. 3. The aggravation of colitis after FMT was a consequence of the whole gut microbiota, not the transfer of Pg only. (a) Diagram of microbiota transplantation experiment. ABX: antibiotic cocktail. FMT: fecal microorganism transplantation. Pg[#], the actual content of Pg in the feces of DSS+LiPg group mice. Transplantation of the Pg[#]/fecal suspension of Nor+Pg[#]/feces of DSS+Pg group into recipient mice→the FMT(Pg[#]/Nor+Pg[#]/DSS+Pg) group. (b) Body weight of FMT(Pg[#]), FMT(Nor+Pg[#]), and FMT(DSS+Pg) mice during DSS-induced colitis, n=5. (c-g) Disease Activity Index (DAI) (c), colon length (d-e), and histological score (f-g) of three group mice on day 8 after DSS induction, n=5. Scale bar, 200 μm. Data are presented as the mean ± SEM; * $p < 0.05$; ** $p < 0.01$; *** $p < 0.001$ by one-way ANOVA. Pg, *Porphyromonas gingivalis*.

6. The results of the microbial analysis of the feces of Pg challenged versus non challenged mice report a significant elevation in the *Bacteroidetes* phylum in challenged animals. How much of this increase was a consequence of the presence of Pg-itself a member of the *Bacteroidetes*?

R: We thank the reviewer for this rigorous question. Pg itself is a member of the phylum *Bacteroidetes*. Tracing the metagenomic sequencing results of the mice feces from the DSS+PBS group, DSS+LiPg group, and DSS+DePg group, the numbers of Pg were estimated to be 4 to 5 log₁₀ units lower than the total gut bacterial counts (Response-Fig. 4a) and to be 4 log₁₀ units lower than the phylum *Bacteroidetes* bacterial counts (Response-Fig. 4b). Besides, the ratio of Pg to the phylum *Bacteroidetes* significantly increased in the DSS+LiPg group compared with the DSS+PBS group, but there was no statistical difference between the DSS+DePg group and the DSS+PBS group (Response-Fig. 4b, $p > 0.05$). The above data indicated that the increase of Pg made minimal contributions to the increase of the phylum *Bacteroidetes*. We have supplemented this result of the numbers of Pg as Supplementary Fig. 6c.

[Figure redacted]

Response-Fig. 4. The abundance of Pg in the mice feces detected by the metagenomic sequencing. (a) The box plots indicate the relative abundance of the Pg species in the mice feces

from the DSS+PBS group, DSS+LiPg group, and DSS+DePg group. **(b)** The box plots indicate the ratio of the Pg to the phylum *Bacteroidetes* in the corresponding groups. Pg, *Porphyromonas gingivalis*.

6. Did the metabolomic assays determine the actual change in concentration of LA in the intestines of the differently treated animals. This is relevant to Point 2 above but also to the interpretation of the dose response in vitro experiments which indicate that 50uM was an important threshold.

R: We thank the reviewer for this rigorous question. Here, we used gas chromatography-mass spectrometry (GC-MS) to quantitatively detect the actual changes in concentration of LA in feces and colon tissues in the DSS+LiPg and DSS+PBS groups. The results showed that the LA concentrations in the feces and colon tissues of DSS+LiPg group were significantly lower than those of DSS+PBS mice (Response-Fig. 5a-b, $p=0.0377$, feces; $p=0.0345$, colon). The average concentration of LA in the intestine of DSS mice receiving Pg gavage stimulation was about equal to 350 μM . Previous studies showed that the highest concentration of non-toxic LA was 200 μM to the CD4^+ T cells *in vitro*[5]. Therefore, in this article, we set a concentration gradient of 0, 25, 50, 100, 200 μM to investigate the effect of LA on IL-17 expression under Th17 inducing conditions *in vitro* and found that 50 μM was an important threshold.

[Figure redacted]

Response-Fig. 5. The actual change in concentration of linoleic acid (LA) in feces and colon tissues of DSS colitis mice after Pg challenge. (a) Quantitative detection of LA in intestinal feces of the DSS+PBS group and the DSS+LiPg group mice by gas chromatography-mass spectrometry (GC-MS). (b) Quantitative detection of LA in the colon tissues of these two groups mice.

Reviewer #2:

In this manuscript, Jia et al demonstrate the protective effect of Linoleic acid (LA) in colitis. Porphyromonas gingivalis (Pg) acted as a critical pathogen in colitis and Pg-induced gut dysbiosis led to decreased LA production. LA administration ameliorates colitis in DSS model through controlling Th17/Treg balance (decrease Th17, increase Treg population). In the details, LA suppressed Th17 population through regulating STAT1 activation via AHR. This is a straightforward study that demonstrate protective effect of LA in colitis using in vitro and in vivo system. The conclusions are largely supported by the presented data and the experimental design is overall sound. However, there is an issue that need to be clarified before the manuscript can be considered any further:

R: We appreciate the reviewer's comments, the reviewer's thoughtful suggestions are valid and well taken.

1. In many facs plots, gating is not correct. For example, in Figure 1J, gating does not include all IL-17 positive population.

R: We apologize for the incorrect gating. We have rechecked the FACS gating in all figures and modified the correct plots, including Fig. 1h and 1J, Fig.2h and 2j.

2. In Supplementary Fig 6, how to analysis cell proliferation? Which method did you use? Authors should mention that and explain in method section.

R: We appreciate this useful suggestion. We used carboxyfluorescein succinimidyl ester (CFSE) labeling combined with ModFit LT™ software to analyze cell proliferation statistically. The specific methods were that LA was added to the activated CD4⁺ T cells or induced Th17 or Treg cells culture system at final concentrations of 0, 25, 50, 100, and 200 μM. Parental CD4⁺ T cells were labeled with CFSE by a CellTrace™ CFSE Cell Proliferation Kit (Invitrogen) according to the manufacturer's instructions; then, different concentrations of LA were added. Flow cytometry collected and detected the CD4⁺ T cells three days later. The obtained data were analyzed by ModFit LT™ software, and the proliferation index was statistically analyzed. We supplemented and revised the manuscript in the Method section (Lines 618-626).

3. In Supplementary Fig 7, how to analysis apoptosis? Which method did you use? Authors should mention that and explain in method section. And X and Y axis labeling is missing.

R: We appreciate this useful suggestion. We used flow cytometry combined with Annexin V-PI Apoptosis Kit (BD Pharmingen) to detect the percentage of cell apoptosis. The specific methods were that LA was added to the activated CD4⁺ T cells or induced Th17 or Treg cells culture system at final concentrations of 0, 25, 50, 100, and 200 μM. The Annexin V-PI Apoptosis Kit (BD Pharmingen) and flow cytometry were used to detect the apoptosis rate of CD4⁺ T cells in each group according to the manufacturer's protocols. Statistical analysis of early apoptosis, late apoptosis, and total apoptosis rates were analyzed by FlowJo software. We have supplemented and revised the manuscript in the Method section (Lines 627-629). We have labeled X and Y axis in revised Supplementary Fig. 10.

4. If authors show gating strategy for FACS analysis, it will be more informative. Especially, analysis for MLN and LPL

R: We appreciate this suggestion. We have complemented the gating strategies for the FACS analysis in our experiments, especially for the detection of the percentages of IL-17⁺Th17 cells (Response-Fig. 6a) and Foxp3⁺Treg cells (Response-Fig. 6b) from MLNs and LPLs as the following. These data were added into the revised Supplementary information as the Supplementary Fig. 14.

[Figure redacted]

Response-Fig. 6. The gating strategies for FACS analysis for the detection of Th17 and Treg.
(a) The gating strategies for the FACS analysis of the percentages of IL-17⁺Th17 cells in MLNs

and LPLs. (b) The gating strategies for the FACS analysis of the percentages of Foxp3⁺Treg cells in MLNs and LPLs.

5. Authors used different concentration of DSS (2%, 3%). Is there any reason?

R: We thank the reviewer for the thoughtful question. As mentioned in the Methods section, mice were administered 2-3% (w/v) DSS in their drinking water ad libitum for seven days to induce acute experimental colitis. The DSS-induced colitis model is widely used because of its simplicity and similarities with human ulcerative colitis. Acute, chronic, and relapsing models of intestinal inflammation can be achieved by modifying the DSS concentration and administration frequency[6]. Therefore, we used 3% DSS for acute inflammation induction in normal mice. Still, when the antibiotic cocktails were administered for gut microbiota depletion, we reduced the concentration to 2% to prevent pre-antibiotic-treated mice or KO mice from experiencing enhanced sensitivity to DSS[7]. We have revised the manuscript's Methods section to clarify this part (Lines 541-546, References 66).

6. In Figure 6, the expression of STAT5 increased by AHR antagonist and no changed by LA treatment. What is the different MOA of LA/AHR between STAT1 and STAT5?

R: We thank the reviewer for the thoughtful question. We again used AHR knockout mice to validate our experimental results. The WB experiment results on the selective interaction between LA/AHR and the Stat family showed that only Stat1 phosphorylation at the Ser727 site was the downstream reaction after LA and AHR binding, rather than Stat5 or Stat3 (Response-Fig. 7a-f). The development of Th17 cells is regulated negatively by IFN- γ , IL-27, and IL-2, the signals of which are dependent on Stat1 (IFN- γ and IL-27) and Stat5 (IL-2), respectively[8-10]. We used AHR antagonist-CH223191 before, and the previous results showed that the expression of Stat5 increased by CH223191 and no change by LA treatment, and the expression of Stat1 increased by LA treatment and no change by CH223191. The possible reason was that LA and CH223191 mediate Stat1 and Stat5 signaling pathways, respectively. We have added these new WB data using AHR KO mice as modified Fig. 6 d-i and revised the manuscript correspondingly in Lines 316-321.

[Figure redacted]

Response-Fig. 7. Only Stat1 phosphorylation at the Ser727 site was the downstream reaction after LA and AHR binding. (a-f) Stat1, Stat3, Stat5, and phosphorylated proteins at different sites were detected by Western Blotting after adding LA into the Th17-inducing culture conditions isolated from AHR WT and KO splenocytes.

Reviewer #3:

In this paper, the authors have elucidated the causative relationship between periodontitis and colitis and demonstrated that the periodontal pathogen-*Porphyromonus gingivalis* (P.g.) aggravates colitis by increasing the abundance of Bacteroides that reduces the level of linoleic acid. Linoleic acid binding to AHR results in the increased phosphorylation of Stat1, leading to the decreased ratio of Th17/Treg. The authors presented a detailed molecular mechanism to clarify the association between periodontitis and colitis; however, additional data and clarification are needed to improve the manuscript and strengthen the conclusion.

Furthermore, the manuscript has many mistakes; the authors should prepare it more carefully before submission.

R: We appreciate the reviewer's comments. We have thoroughly considered all their points and performed additional experiments to answer their questions.

Major issues:

Experimental design:

The authors have administrated P.g. by oral gavage to investigate the pathological relevance between periodontitis and IBD. This method allows them to analyze the effect of P.g. on colitis, but not suitable for determining the pathological relevance of periodontitis. To rigorously examine it, they should orally infect mice with P.g. to develop periodontitis and then investigate susceptibility to experimental colitis.

R: We thank the reviewer for this thoughtful suggestion. We designed the following experiment for further investigation (Response-Fig. 8a). The degree of alveolar bone resorption was evaluated by measuring the height from the cemental-enamel junction (CEJ) to the alveolar bone crest (ABC). The alveolar bone resorption in the DSS+PD group was significantly severer than that in the DSS+PBS group and the DSS+Pg group, indicating that the PD model was successfully constructed (Response-Fig. 8b-c). The DSS+Pg group mice developed more severe intestinal inflammation than the DSS+PD and the DSS+PBS ones based on body weight, DAI, colon length, HAI (Response-Fig. 8d-i, $p < 0.001$). Moreover, the colitis in the DSS+PD group was also significantly more severe than that in the DSS+PBS group (Response-Fig. 8d-i, $p < 0.05$), proving that periodontal tissue inflammation enhanced the susceptibility to experimental colitis.

[Figure redacted]

Response-Fig. 8. Periodontal tissue inflammation enhanced the susceptibility to experimental colitis. (a) The experimental design. Silk ligation and local infection of Pg to establish the periodontitis (PD) model. WT mice were gavaged with PBS or Pg for 14 days with the last seven days of treatment with 3.0% DSS, named with DSS+PBS group, DSS+PD group, and DSS+Pg group. (b-c) Micro-CT scanning showed significant alveolar bone loss at the maxillary second molars in the DSS+PD group. The distance from the CEJ to the ABC showed the degree of alveolar bone resorption. $n=5$. (d) Body weight of DSS+PBS, DSS+PD, DSS+Pg mice during DSS-induced colitis, $n=5$. (e-i) DAI, colon length, and HAI of DSS+PBS, DSS+PD, DSS+Pg mice on day 8, $n=5$. Scale bar, 200 μm . Data are presented as mean \pm SEM; * $p < 0.05$; ** $p < 0.01$; *** $p < 0.001$ by one-way ANOVA test. Pg, *Porphyromonas gingivalis*. CEJ, cemental-enamel junction. ABC, alveolar bone crest. DAI, disease activity index. HAI, histological activity index.

Introduction

L.75-76: The term “metabolic disorder” represents "illness" related to the metabolic system, such as diabetes and obesity. However, the authors deem this term to indicate the alteration in microbial metabolism caused by the altered gut microbial composition, which may mislead the readers.

R: We appreciate this comment and suggestion. We have rechecked the manuscript and modified the term “metabolic disorder” to “metabolic regulation” (Line 86 and 233).

Moreover, the metabolite alteration is poorly mentioned in the introduction. The authors should provide further evidence on the association between metabolite alteration and periodontitis/IBD.

R: We appreciate this comment and suggestion. We have added more corresponding references on the association between metabolite alteration and periodontitis/IBD (Lines 73-83, references 18-22).

Result

1.In Fig.1, It remains obscure whether P.g. colonizes and expands in the intestinal lumen. The authors should show the abundance of P.g. in the intestine after seven days of oral gavage of P.g.

R: We thank the reviewer for this thoughtful suggestion. We detected the Pg content by using qPCR in the feces of the DSS+Pg group at different times (please see Response-Fig. 2). The Pg content in the feces reached its peak on the 3rd day after receiving Pg (Response-Fig. 2b, Day 3, $2.14 \times 10^9 \pm 3.29 \times 10^8$) and began to decrease from the 5th day (Day 5, $1.27 \times 10^8 \pm 2.28 \times 10^7$). When DSS was started, the actual detectable level of Pg in the intestine was already low (Day 7, about $9.19 \times 10^7 \pm 3.77 \times 10^7$). Till the 14th day, the Pg content decreased to $2.65 \times 10^8 \pm 1.16 \times 10^8$, though higher than the 7th day, there was no statistical significance (Day 7 vs. Day 14, $p > 0.05$).

2.In Fig. 11-n, the authors have claimed that the exacerbation of colitis by gavage of P.g. depends on the increased ratio of Th17/Treg. However, the increased Th17/Treg ratio may be a consequence, but not a cause, of the exacerbation of colitis, given that they analyzed it after the development of colitis (Day 7). The authors should reanalyze Th17/Treg balance before the onset of colitis (e.g., Day 2). Moreover, although it is controversial, DSS-induced colitis does not depend on Th17 cells (DOI: 10.4110/in.2011.11.6.416). Activation of macrophage/monocytes plays a vital role in the pathogenesis of this colitis model. Therefore, the authors should carefully analyze myeloid cell subsets, including monocytes, CX3CR1low (inflammatory), and CX3CR1high (anti-inflammatory) macrophages at the early stage of inflammation.

R: We appreciate this comment and suggestion. We checked the Th17/Treg balance and CX3CR1⁺ macrophages in the colon at the early stage of colitis (Day 2) after gavage of Pg (Please see Response-Fig. 9a). WT mice were given the antibiotic cocktails for gut microbiota depletion before DSS treatment (ABX). Then, mice were gavaged with Pg for nine days, with the last two days of treatment with 3.0% DSS. DSS+Pg group mice didn't show more severe inflammation than the ABX(DSS+Pg) ones based on body weight, DAI, and HAI (Response-Fig. 9b, c, f, g, $p > 0.05$), except for the difference in the colon length (Response-Fig. 9d, e, $p=0.0463$). The percentages of IL-17⁺Th17 of the DSS+Pg group in MLNs and LPLs was significantly higher, and the percentages

of Foxp3⁺Treg of the DSS+Pg group in MLNs and LPLs was significantly lower compared with the ABX(DSS+Pg) group (Response-Fig. 9h-m, Th17, DSS+Pg vs. ABX(DSS+Pg), $p<0.01$; Treg, DSS+Pg vs. ABX(DSS+Pg), $p<0.01$). The Th17/Treg ratios in MLNs and LPLs in the DSS+Pg group were also significantly higher than that in the ABX(DSS+Pg) group (Response-Fig. 9n), indicating that the Th17/Treg imbalance occurred in the early stage of colonic inflammation after gavage of Pg in a gut microbiota-dependent manner. We also checked the macrophages (Mφs) in the colonic lamina propria (LPM), the percentage of the CX3CR1⁺ macrophages from the CD45⁺F4/80⁺Ly6C⁺ leukocytes in the DSS+Pg group was significantly lower than that in the ABX(DSS+Pg) group (Response-Fig. 9o-p, $p<0.01$).

A previous report showed that CX3CR1-KO mice had a more severe disease since the early beginning of the DSS-induced colitis[11], which was consistent with our results. The behaviors of gut macrophages lacking the receptor CX3CR1 during inflammation have been investigated in some studies with controversial results. Other studies demonstrated that CX3CR1⁺ macrophages elicited a pro-inflammatory response in DSS colitis by producing iNOS[12]. On the other hand, CD4⁺ T lymphocytes are crucial in the pathogenesis of IBD[13], and Th17 combined phenotypes of CD4⁺ T cells are found accumulated in the intestinal mucosa of IBD patients[14]. Taking together, we believed that both Th17/Treg and macrophages played important roles in the pathogenesis of this colitis model. We added these data as revised Supplementary Fig. 5. and made additional discussion about this in our revised manuscript (Lines 437-442, References 43-45).

[Figure redacted]

Response-Fig. 9. Increased Th17/Treg ratio happened at the early stage during the exacerbation of colitis by gavage of Pg. (a) Diagram of this experiment. WT mice were administrated with antibiotics cocktail for five days to gut microbiota depletion before Pg and DSS treatment named the DSS+Pg group and the ABX(DSS+Pg) group. Mice were gavaged with Pg for nine days, with the last two days of treatment with 3.0% DSS. (b) Body weight of the DSS+Pg group and ABX(DSS+Pg) group mice during DSS-induced colitis, $n=5$. (c-g) DAI, colon length, and HAI of the DSS+Pg group and ABX(DSS+Pg) group mice on day 2 after DSS induction, $n=5$. Scale bar, 200 μm . (h-k) The proportions of IL-17⁺Th17 cells, Foxp3⁺Treg cells, and their ratio among total CD4⁺T cells within MLN and LPL were detected by flow cytometry and statistically analyzed (l-n), $n=5$. (o) Phenotype of CX3CR1⁺ macrophages in the colonic lamina propria by FACS analyses. (p) The percentage of the CX3CR1⁺ macrophages were calculated relative to all CD45⁺F4/80⁺Ly6C⁺ cells. Data are presented as the mean \pm SEM; * $p<0.05$; ** $p<0.01$; *** $p<0.001$ by one-way ANOVA. Pg, *Porphyromonas gingivalis*; MLN, mesenteric lymph nodes; LPL, lamina propria lymphocytes; LPM, colonic lamina propria macrophages.

3.In Fig. 2, the authors have stated that the exacerbation of DSS-induced colitis and Th17/Treg imbalance are attributed to the alteration of gut microbiota. Nevertheless, the FMT experiment does not exclude the possibility that P.g. directly influences the

inflammatory status and T-cell balance, considering that donor feces are likely to contain P.g. (Fig. 3g).

R: We appreciate this comment and suggestion. We conducted the new experiments to support that the severity of colitis was due to the altered gut microbiota composition instead of the power of Pg itself (please see Response-Fig. 3a). Firstly, the content of Pg in the feces of DSS+LiPg group mice was quantitatively detected by qPCR and labeled as Pg[#], which was about $2.65 \times 10^8/g$ in the feces at the endpoint of modeling. The same Pg[#] bacterial load were added to normal mouse fecal suspension and labeled as Nor+Pg[#]. The gut microbiota-depleted recipient mice (GDR mice) were reconstituted with equal volumes of Pg[#] liquid (FMT(Pg[#]) group), or Nor+Pg[#] (FMT(Nor+Pg[#]) group), or the fecal suspension from DSS+LiPg-treated mice (FMT (DSS+Pg) group) via intragastric administration. FMT(DSS+Pg) group mice developed more severe inflammation than FMT(Pg[#]) group and FMT(Nor+Pg[#]) group as judged by weight loss, DAI, colon length, HAI (Response-Fig. 3b-g), suggesting that the enhanced colitis in GDR mice were caused by changes in gut microbiota through consumption of feces from the DSS+LiPg mice. Moreover, FMT(Pg[#]) group mice developed almost the same degree of inflammation compared with FMT(Nor+Pg[#]) ones (Response-Fig. 3b-g, $p > 0.05$), indicating that Pg in the FMT was not the dominant factor leading to intestinal pathology. We added these data as revised Supplementary Fig. 4c-i and revised the manuscript correspondingly in Lines 160-164.

4.In Fig2A, the authors should show gut microbial composition after the FMT to confirm the reconstruction of donor microbiota.

R: We thank the reviewer for the professional question. We detected the fecal bacterial genomes from the DSS+LiPg group and FMT(DSS+LiPg) group mice by metagenomic sequencing to confirm the reconstruction of donor microbiota (Response-Fig. 10). Common species analysis (Response-Fig. 10a) showed 327 common species between the two groups. 209 species were special for the DSS+LiPg group and 384 species were special for the FMT(DSS+LiPg) group. Diversity index analysis (Response-Fig. 10b, PCoA graph) showed that the samples from FMT(DSS+LiPg) group had similarity with ones from the DSS+LiPg group. The permutational multivariate analysis of variance (PERMANOVA) showed no significant difference between the two groups ($p=0.1$). Bacterial composition analysis (Response-Fig. 10c) at the species level showed that among the top 20 bacterial species with the highest abundance in all samples, the most significant different species was *Duncanella Dubois*, which belonged to the phylum *Bacteroidetes*. Bacterial composition analysis (Response-Fig. 10d) at the phylum level showed that the top 4 phyla with the highest abundance were *Bacteroidetes*, *Verrucomicrobia*, *Firmicutes*, and *Actinobacteria*. Among them, *Bacteroidetes* increased, and *Firmicutes* decreased significantly in the FMT(DSS+LiPg) group mice compared with the DSS+LiPg group (Response-Fig. 10e-h). Phyla significance difference analysis based on LEfSe (Response-Fig. 10i) showed that when $LDA > 4$, the significant difference in the FMT(DSS+LiPg) group was *Bacteroidetes*. We added these data and modified as Supplementary Fig. 3 in the revised manuscript (Lines 138-139).

[Figure redacted]

Response-Fig. 10. The reconstruction of donor microbiota after FMT. (a) High-throughput metagenomic sequencing of the faecal bacterial genomes from DSS+LiPg and FMT (DSS+LiPg)

mice. n=3. The number of common or unique species between the two groups in the Venn diagram. (b) 3D PCoA graph based on the Bray–Curtis distance matrix. PERMANOVA: F=13.895, P value=0.1. PERMANOVA: permutational multivariate analysis of variance. (c) Column diagram of the relative distribution of each sample at the species level (top 20). (d) Column diagram of the relative distribution of each sample at the phylum level (top 10). The Y-axis is the sequence number percent, indicating the ratio of this phylum level to the total annotation data. (e-h) The box plots indicate the top 4 relative abundances of each bacterial group at the phylum level (*Bacteroidetes*, *Firmicutes*, *Verrucomicrobia*, and *Actinobacteria*). (i) The most differentially abundant taxa of characteristic microorganisms at the phylum level between the DSS+LiPg and FMT (DSS+LiPg) group mice based on ANCOM (LDA Score=4 by LEfSe). LDA Score: linear discriminant analysis score. ANCOM: analysis of composition of microbiomes.

5. In Fig. 3c, the gut microbial composition should be shown not only at the phylum level but also at the species level to provide the relative abundance of P.g..

R: We thank the reviewer for this detailed suggestion. Here we showed the gut microbiota composition at the species level (Top 20 species with relative abundance) (Response-Fig. 11a). The top 4 species with the most relative abundance were *Bacteroides caecimuris*, *Akkermansia muciniphila*, *Faecalibaculum rodentium*, and *Bacteroides cellulosilyticus*, corresponding to the representative species with the most significant differences in the DSS+LiPg, DSS+PBS, and DSS+DePg groups after the LEfSe analysis results (Fig. 3f and Supplementary Fig. 4e-h in the revised manuscript), respectively. The species *Porphyromonas gingivalis* were significantly higher in the DSS+LiPg group and the DSS+DePg group than in the DSS+PBS group (Response-Fig. 11b). The numbers of Pg were estimated to be 4 to 5 log₁₀ units lower than the total gut bacterial counts and ranked about 170th in the relative abundance of all fecal microbiota. We have added the box plots of *Porphyromonas gingivalis* abundance as the Supplementary Fig. 6c.

[Figure redacted]

Response-Fig. 11. The gut microbial composition at the species level. (a) Column diagram of the relative distribution of each sample on the species level (Top 20). The Y-axis is the sequence number percent, indicating the ratio of this species level to the total annotation data. (b) The box plots of the *Porphyromonas gingivalis* (Pg) abundance in each group.

6. In Fig 3F, it is an overstatement that Bacteroides are responsible for the exacerbation of colitis. To conclude this, the authors need to perform experiments using gnotobiotic mice associated with AfAas2 knocked-out Bacteroides caecimuris.

R: We thank the reviewer for this detailed suggestion. After we analyzed the correlation between microbial abundance and environmental factors (HAI, DAI, colon length, and weight), we found a significant relationship between the gut microbiota phyla and the HAI, DAI, colon length, and weight in colitis mice (Response-Fig. 12a, overall permutation test p=0.0011). There was a positive correlation between the phylum *Bacteroidetes* and HAI/DAI while a negative correlation with colon length and body weight (Response-Fig. 12a, b), which meant the increase of the *Bacteroidetes* abundance was positively correlated with the exacerbation of colitis after gavage of

Pg. The phylum *Firmicutes* was negatively correlated with HAI and DAI but positively correlated with colon length and body weight (Response-Fig. 12a, b), consistent with the abundance of *Firmicutes* decreased following the gastric administration of Pg exacerbating colitis. What's more, the angle between the *Bacteroidetes* and HAI was significantly smaller than the angle between the *Firmicutes* and the extension line of HAI, indicating that the phylum *Bacteroidetes* has a stronger correlation with environmental factors than the phylum *Firmicutes*. The correlation heat maps also reflected that the phylum *Bacteroidetes* has the strongest correlation with environmental factors (Response-Fig. 12b).

Based on these results, we believe that the aggravation of DSS colitis caused by the gavage of Pg is strongly related to the expansion of *Bacteroidetes*. We have supplemented these correlation analysis data in the revised manuscript as Supplementary Fig. 6i, j in Lines 194-209. We have revised the manuscript about the description of Fig.3f in Lines 213-217 as "The data above indicated that the reason why the colitis in DSS+LiPg group and DSS+DePg group was more severe than that in DSS+PBS group probably due to the more unbalanced gut microbiota composition characterized by the increase of *Bacteroidetes* (dominated) and the decrease of *Firmicutes*." to avoid the overstatement that *Bacteroidetes* are responsible for the exacerbation of colitis.

[Figure redacted]

Response-Fig. 12. The correlation analysis between microbial abundance and environmental factors (HAI, DAI, colon length, and weight). (a) RDA ranking diagram at the phylum level. Arrows represent environmental factors, and the length of the arrow line represents the degree of correlation between the environmental factor, community distribution, and species distribution (explaining the magnitude of variance). The longer the arrow, the more significant the correlation. Each point represents a phylum; the more significant the point, the higher the phylum's abundance. The angle between the arrow line and the point represents the correlation between a specific environmental factor and the phylum. The smaller the angle, the higher the correlation. Acute angle: positive correlation; Obtuse angle: negative correlation; Right angle: no correlation. (b) The correlation heat maps between the phyla and environmental factors. The X-axis represents environmental factors, while the Y-axis represents phyla. Red represents a positive correlation, while blue represents a negative correlation. $*0.01 \leq p < 0.05$; $**0.001 \leq p < 0.01$; $***p < 0.001$. RDA, redundancy analysis.

7. In Fig. 3g, Because Pg is also detected in the DePg group, there are non-specific signals. The authors should reanalyze it using more specific probes or at least explain it.

R: We thank the reviewer for this detailed question. Here the Pg ultrasonicate used was prepared from whole bacterial cells as described in the Methods section. Because the gavage was performed every other day throughout the experiment, we treated the DSS+DePg group mice with Pg extract just one day before the animals were euthanized. Therefore, it was possible to detect the Pg DNA signal in the colon tissue of the DSS+DePg group mice via the Pg-DNA probe, considering the heterogeneous nature of the extract.

8. In L. 148-149 (Fig. 3b), the authors have described that the samples from the DSS+LiPg group show lower similarity than those from the DSS+DePg group in species composition

structure compared with those from the control group. To discuss the similarity of microbial structure among the groups, they should quantify the distance between the samples and show the re in a box plot.

R: We appreciate the reviewer's professional suggestion. We have shown the alpha diversity analysis of the fecal bacteria from the DSS+PBS, DSS+LiPg, and DSS+DePg group mice (please see Response-Fig. 1). The Chao1 index calculating the community richness based on the number of OTUs showed no significant difference among the three groups (Response-Fig. 1a-c, $p > 0.05$). The Simpson index calculating the community diversity showed a significant difference among the three groups (Response-Fig. 1d-f, $p=0.027$). The p-value of the DSS+LiPg group *versus* the DSS+PBS group ($p=0.039$) was smaller than that of DSS+DePg *versus* DSS+PBS ($p=0.049$), indicating that the samples from the DSS+LiPg group showed lower similarity than those from the DSS+DePg group in species composition structure compared with those from the control group as described in the manuscript (Fig.3b).

9.L. 165-168, 174-175. the authors have stated that the severity of colitis may be due to the more unbalanced gut microbiota composition and the significant increase of Bacteroides. However, as pointed out above, the possibility that Pg itself may exacerbate colitis has not been excluded. It is not sure that the degree of colitis is related to the expansion of Bacteroides.

R: We appreciate the reviewer's comments and professional suggestions. We conducted the FMT experiments (please see Response-Fig. 3) to support that the severity of colitis was due to the more unbalanced gut microbiota composition rather than Pg itself. Besides, as we answered in the above Question 6 (please see Response-Fig. 12), the correlation analysis showed that the phylum *Bacteroidetes* had the highest correlation with environmental factors (DAI, HAI, colon length, and body weight). Moreover, the abundance of the *Bacteroidetes* occupied an absolute advantage in the whole gut microbiota (Fig. 3c-e and Supplementary Fig. 6a-b). We believe that the aggravation of DSS colitis caused by the gavage of Pg is strongly related to the expansion of *Bacteroidetes*. We have added these data of the correlation analysis in Response-Fig. 12 and modified as Supplementary Fig. 6i, j to support our conclusion.

10. Even so, it is uncertain why the authors focused on metabolites. The information “Studies have shown that all members of the Bacteroides phylum express AfAas2, which is essential for the metabolism of long-chain fatty acids” (lines 175-177) should be provided before the hypothesis (lines 173-175).

R: We appreciate the reviewer's comments and questions. Gut microbes and their interactions with hosts are crucial in IBD pathogenesis and progression[15]. One of the critical mechanisms by which the gut microbiota interacts with the host is through metabolites, which are small molecules produced as intermediate or end products of microbial metabolism[16]. The host-microbe metabolic axes act upon multiple organ systems to modulate the host's physiology[17]. Studying the complex physiological systems during the disease assessment often overwhelmed the single-omic capabilities. With the increasing application of metagenomics and metabolomics in studying the pathogenesis of IBD[18], we focus on the changes in the gut microbiota and its related metabolic pathways to study the mechanism of the aggravation of colitis after gavage of Pg via the “gum-gut” axis. We have revised the manuscript and supplemented relevant references in Lines 220-233 (References 23-25) to help readers better understand why we focused on metabolite

pathways. We placed “Studies have shown that all members of the Bacteroides phylum express AfAas2, which is essential for the metabolism of long-chain fatty acids” before the hypothesis.

11. In Fig. 5, do the LA concentrations in the in vitro experiments reflect physiological amounts? Please provide the rationale for how the authors determined the concentrations.

R: We thank the reviewer for this rigorous question. Here, we used gas chromatography-mass spectrometry (GC-MS) to quantitatively detect the actual changes in concentration of LA in feces and colon tissues in the DSS+LiPg and DSS+PBS groups (please see Response-Fig. 5). The results showed that the LA concentrations in the feces and colon tissues of DSS+LiPg group were significantly lower than those of DSS+PBS mice (Response-Fig. 5a-b, $p=0.0377$, feces; $p=0.0345$, colon). The average concentration of LA in the intestine of DSS mice receiving Pg gavage stimulation was about equal to 350 μM . Previous studies showed that the highest concentration of non-toxic LA was 200 μM to the CD4⁺ T cells *in vitro*[5]. Therefore, in this article, we set a concentration gradient of 0, 25, 50, 100, 200 μM to investigate the effect of LA on IL-17 expression under Th17 inducing conditions *in vitro* and found that 50 μM was an important threshold.

12.L. 248-252, this statement is not convincing because the protein expression levels are comparable between the absence or presence of LA (Fig. 6f, h, g, i).

R: We thank the reviewer for the comments and this thoughtful question. We completely agree that the statement “we used RT-qPCR to examine Stat1, Stat3, Stat5 expression in the con+LA group under Th17-polarizing conditions and found that LA promoted the expression of Stat1, Stat5 and inhibited the expression of Stat3 (Fig. 6a-c), indicating that LA-AHR complex may negatively regulate the generation of Th17 cells by modifying the activation of Stat1 and Stat5.” of the original manuscript was not very rigorous since we cannot rely solely on RT-qPCR results to summarize the relationship between LA and the Stat family proteins in regulating Th17 cell differentiation. We have omitted the sentence “indicating that LA-AHR complex may negatively regulate the generation of Th17 cells by modifying the activation of Stat1 and Stat5”. We again used AHR knockout mice to validate our experimental results. The WB experiment results on the selective interaction between LA/AHR and the Stat family showed that only Stat1 phosphorylation at the Ser727 site was the downstream reaction after LA and AHR binding, rather than Stat5 or Stat3 (please see Response-Fig. 7). We have added these new WB data using AHR KO mice as modified Fig. 6 d-i and revised the manuscript correspondingly in Lines 316-321.

13. Fig. 7, To strengthen the authors’ conclusion, T-cell-specific Ahr KO mice such as Lck-Cre Ahr^{flox/flox} mice (doi: 10.1073/pnas.1111786108; 10.1073/pnas.0504757102) should be employed. Considering that various cell types, including dendritic cells and intestinal epithelial cells, express Ahr (doi: 10.1016/j.immuni.2017.12.012, 10.3389/fimmu.2021.638725), germline KO of Ahr may exert considerable influence on these cell types to affect Th17/Treg balance indirectly.

R: We thank the reviewer for this thoughtful question. Considering that we currently do not have readily available Lck-Cre Ahr^{flox/flox} mice and the long obtaining time, we adopt another method to achieve T-cell-specific Ahr KO mice. The experimental plan was as follows (Response-Fig. 13a). Firstly, Naïve CD4⁺ T cells were abstracted from spleen cells of wild-type (WT) mice and AHR knockout mice. Then 2×10^6 cells were injected into Rag2 KO mice via tail vein to construct experimental mice containing targeted knockout of AHR from the T cells, named Control mice

and T-AHR-KO mice, respectively. Then the two groups of mice were gavaged with Pg every other day for 14 consecutive days, and DSS was administered starting from the last seven days.

The inflammation of colitis in the T-AHR-KO(DSS+Pg) group mice was significantly lighter than that in the Control(DSS+Pg) group mice as evidenced by the weight loss, DAI, the colon length and HAI (Response-Fig. 13b-g, $p < 0.01$), confirming that gavage of Pg aggravating the colitis was dependent on AHR of T cells. Consistent with this, the T-AHR-KO(DSS+Pg) group showed decreased percentages of IL-17⁺Th17, higher percentages of Foxp3⁺Treg, and significantly lower Th17/Treg ratios in MLN and LPL compared with the Control(DSS+Pg) group (Response-Fig. 13h-n, $p < 0.01$), further demonstrating that AHR of T cells mediated the Th17/Treg imbalance induced by intragastric Pg gavage to colitis.

[Figure redacted]

Response-Fig. 13. AHR of T cells mediated the Th17/Treg imbalance induced by intragastric Pg gavage to colitis. (a) The experimental design. Naïve CD4⁺ T cells were abstracted from spleen cells of wild-type (WT) mice and AHR knockout mice. Then 2×10^6 cells were injected into Rag2 KO mice via tail vein to construct experimental mice containing targeted knockout of AHR on the T cells surface, named Control mice and T-AHR-KO mice, respectively. Before DSS induced colitis model was established, Control or T-AHR-KO mice were administered with live Pg and named with Control(DSS+Pg) and T-AHR-KO(DSS+Pg), respectively. (b-c) Body weight (b) and disease activity index (DAI) (c) of mice during DSS-induced colitis, $n=5$. (d-e) Colon length of these two groups of mice on day 8 after DSS induction, $n=5$. (f-g) Histological score of these two groups of mice. Scale bar, 200 μm . (h-n) The proportions of IL-17⁺Th17 cells, Foxp3⁺Treg cells, and their ratio among total CD4⁺ T cells within MLN and LPL were detected by flow cytometry and statistically analyzed. $n=5$. Data are presented as mean \pm SEM. ** $p < 0.01$; *** $p < 0.001$ by one-way ANOVA.

14.Fig.8 indicates that the binding of LA to Ahr facilitates the dissociation of HSP90 and the nuclear translocation of Ahr. However, no experimental evidence supports this model. Also, it remains unclear how LA-bound Ahr increases phosphorylated STAT1.

R: We thank the reviewer for this professional question. We used the WB method to detect the changes of AHR and HSP90 in the cytoplasm and nucleus after adding LA under the induction of Th17 conditions to demonstrate the dissociation of HSP90 and the nuclear translocation of AHR. Here, HSP90 was only detected in the cytoplasm, and its expression was increased in the con+LA group (sample 2); the expression of AHR in the con+LA group decreased in the cytoplasm but increased in the nucleus (Response-Fig. 14a). These results indicated that after LA bound to AHR, the HSP90 and AHR complex dissociated and then AHR entered the nucleus. At the same time, we detected the expression of AHR and ARNT (AHR nuclear translocator) in the nucleus under the non-reduced denaturing conditions (without β -Mercaptoethanol). Both AHR and ARNT were expressed higher in the con+LA group, confirming that AHR bound to ARNT after entering the nucleus (Response-Fig. 14b). AHR is a ligand-activated transcription factor in the cytoplasm, forming a complex with HSP90, a XAP-molecule 2 (XAP2), and P23. Upon binding with a ligand, AHR undergoes a conformation change, translocates to the nucleus, and dimerizes with ARNT. The above results supported this model.

We firstly confirmed that LA was a ligand that inhibited AHR, since the expression of AHR began to be significantly inhibited from the second day after adding LA (Response-Fig. 15a) and different concentrations of LA inhibited the expression of AHR (from 25 μ M to 200 μ M), especially at the concentration of 50 μ M (Response-Fig. 15b). Then we found that only Stat1 phosphorylation at the Ser727 site was the downstream reaction after LA and AHR binding (please see Response-Fig. 7). Besides, AHR negatively regulated Stat1 phosphorylation in the differentiation of Th17 cells[19]. Thus, after the binding of LA and AHR, the phosphorylation of Stat1 was increased, which was confirmed by the FACS results in Fig 6k, n. We added these data and modified Response-Fig. 7 as Fig. 6d-i, added the Response-Fig. 14, 15 as the Supplementary Fig. 12, and revised the manuscript in Lines 316-321 and 303-312.

[Figure redacted]

Response-Fig. 14. The binding of LA to AHR facilitates the dissociation of HSP90 and the nuclear translocation of AHR. (a) WB detected changes in the subcellular translocation of AHR and HSP90 after isolating cytoplasmic and nucleoproteins. (b) The expression of AHR and ARNT in the nucleus was detected under non-reducing conditions after the extraction of nucleoproteins.

[Figure redacted]

Response-Fig. 15. LA inhibited AHR expression under Th17 induction conditions. (a) The expression of AHR after adding LA at different time points in the Th17-inducing culture conditions. (b) The expression of AHR after adding different concentrations of LA on 3rd day under Th17-polarizing conditions.

Discussion

1.In L. 308-309, the authors have mentioned the decreased microbial diversity; however, it is difficult to conclude without data on the alpha diversity of gut microbiota.

R: We thank the reviewer for this detailed suggestion. We have shown the alpha diversity analysis of the fecal bacteria from the DSS+PBS, DSS+LiPg, and DSS+DePg group mice as answered before (please see Response-Fig. 1). The Chao1 index which calculated the community richness based on the number of OTUs showed no significant difference among the three groups (Response-Fig. 1a-c, Kruskal-Wallis's test, $p > 0.05$). The Simpson index which calculated the community diversity showed a significant difference among the three groups (Response-Fig. 1d-f, $p=0.027$). Besides, the Simpson indexes of the DSS+LiPg group and the DSS+DePg group were higher than that of the DSS+PBS group, indicating the DSS+LiPg group and the DSS+DePg group showed the decreased microbial diversity compared with the DSS+PBS group. Thus, we could conclude that “the gut microbiota exhibited decreased microbial diversity after Pg administration” as described in Lines 376-378.

2. In L. 324 and 464, the reference to explain the reason why they reduced the concentration of DSS in the secondary source. Please quote the original citation (doi: 10.1538/expanim.50.387).

R: We apologize for the inaccurate citation. We have revised the manuscript and cited the original literature in Lines 541-546 (Reference 66, doi: 10.1538/expanim.50.387) since it was the original study to determine intestinal microflora's role in DSS-induced colitis and verify the effect of different DSS concentrations on colitis pathological inflammation.

3. In L. 315-324, it seems unnecessary to discuss the methodology, such as the concentration of DSS, because it is out of the focus of the current study. Please omit it or describe it briefly.

R: We appreciate this detailed suggestion. We have removed the discussion about DSS concentration from the Discussion part since we have already explained it in the Methods section (Lines 541-546).

4. In L. 352-354, please cite appropriate references.

R: We appreciate this detailed suggestion. We have cited the appropriate references in Line 444 (References 46-47).

5. In L. 374-378, the data using an Ahr antagonist are not convincing. Quantification of the western blotting data is desirable. It is also possible to speculate that LA may serve as an antagonist for Ahr. Does an Ahr agonist cancel LA-mediated Th17/Treg imbalance?

R: We appreciate this comment and suggestion. We again used AHR knockout mice to validate our experimental results on the selective interaction with the Stat family after LA and AHR binding. (Please see Response-Fig. 7). We found only Stat1 phosphorylation at the Ser727 site was the downstream reaction after LA and AHR binding, rather than Stat5 or Stat3. We have added the quantification of these western blotting data and modified as revised Fig. 6d-i.

We completely agreed that LA might serve as an antagonist for AHR. The expression of AHR began to be significantly inhibited from the second day after adding LA and different concentrations of LA inhibited the expression of AHR (from 25 μ M to 200 μ M) (please see Response-Fig. 15). We have added the above data as revised Supplementary Fig. 12c, d and revised the manuscript in Lines 286-295 and 304-309 of the Results section.

6. The mechanism by which Pg increases Bacteroides is still unclear. The authors should at least discuss it.

R: We appreciate this comment and suggestion. Previous studies revealed that oral administration of Pg induced an elevation of the population belonging to *Bacteroidetes* in the ileal microflora, which coincided with increased systemic inflammation[20]. Another study found that the administration of Pg significantly altered gut microbiota, with an increased proportion of phylum *Bacteroidetes* and a decreased proportion of phylum *Firmicutes*[21]. The *Firmicutes/Bacteroidetes* (F/B) ratio is widely accepted to have an essential influence in maintaining normal intestinal homeostasis since they are the two most critical bacterial phyla in the gastrointestinal tract. Decreased F/B ratio is regarded as dysbiosis, usually observed with IBD[22]. These studies were consistent with our data as we found that the aggravation of colitis

caused by gavage of Pg exhibited characteristics of increased *Bacteroidetes* (dominated) and decreased *Firmicutes*. Usually, Pg was only transiently detected in the gut, where it failed to colonize[20]. The gut colonization by oral pathobionts and promotion of colitis required pre-existing intestinal inflammation (induced by DSS treatment) or a colitis-susceptible host[23]. In our experiments, when DSS was started, the actual detectable level of Pg in the intestine after gavage of Pg was already deficient (estimated to be 4 to 5 log₁₀ units lower than the total gut bacterial counts). In fact, with shallow colonization levels (< 0.01% of the total), Pg triggered changes to the commensal microbiota composition leading to inflammation. However, when homeostasis was compromised, Pg's role became minimal[3, 24]. The above data suggest that the effect of Pg administration differs from other pathogenic bacteria, such as *Salmonella typhimurium*, that usually outgrow indigenous bacteria[21]. Further studies are needed to identify the molecules or metabolites derived from Pg that cause phenotypic and microbiological changes. We have supplemented the discussion about this question in Lines 390-413 (References 30-37).

Minor points (Grammar and Typo)

The manuscript contains numerous grammar errors. English editing by a native speaker is strongly recommended.

R: We apologize for the grammar error. We have polished the entire manuscript to make it more readable and more in line with the publication requirements of the magazine.

The format of the gene name should be in italic throughout the manuscript.

R: We appreciate this detailed suggestion. We have modified the format of the gene name in italic throughout the manuscript.

The abbreviations have double notations (IBD in L43 and L47; Pg in L74 and L84).

R: We apologize for the duplication notations. We have deleted the notation of IBD in Line 74 and the notation of Pg in Line 184 to reduce duplication.

L.55: “RoR γ ” should be “ROR γ .”

R: We appreciate this detailed correction. We have rechecked the manuscript and corrected “RoR γ ” to “ROR γ ” in Lines 56 and 467.

L.53: “T helper cell 17” should be “T helper 17 cell.”

R: We appreciate this detailed correction. We have modified the “T helper cell 17” to “T helper 17 cell” in Line 54.

L.85: “WT” should be spelled out at the first appearance.

R: We appreciate this detailed suggestion. We have spelled out the abbreviation “WT” as “wide type” in Line 99.

L.90: “Supplementary Fig.2a-d” may be “Supplementary Fig.2A-D” following the description in Figures.

R: We appreciate this detailed suggestion. To maintain consistency with the icons in the main text, we have modified all caps marks in the Supplementary Figs to lowercase a, b, c, et al.

Fig1,2 and others; The authors should display the values in a row.

R: We appreciate this detailed suggestion. We have displayed the values in Fig.1,2 in a row.

L.212, 217-218; Please add a space before the units.

R: We appreciate this detailed suggestion. We have rechecked the manuscript and added a space before the units in Lines 268 and 274.

L156,236; “LEFSe” should be “LEfSe.”

R: We apologize for this mistake. We have rechecked the manuscript throughout and modified the word “LEFSe” to “LEfSe” in Lines 188 and 292.

Supplementary Fig6A-C, 7A-C; The authors should enlarge the panels and the axis labels.

R: We appreciate this detailed suggestion. We have enlarged the panels and the axis labels in Supplementary Fig. 9a-c and 10a-c.

L.511,513 “300 g” should be “300×g.”

R: We appreciate this detailed suggestion. We have corrected “300g” to “300×g” in Lines 604 and 609.

L.516; 8 of “1.0×10⁸” should be superscript.

R: We appreciate this detailed suggestion. We have modified “1.0×10⁸” to “1.0×10⁸” in Line 603.

L.572-573; In metagenome analysis, why did you remove the host DNA sequence by aligning it to the human genome (hg38)? The authors should use Mus musculus assembly like mm10.

R: We apologize for this mistake. We used the Mus musculus reference genome (GRCm38.p3) and modified the Methods section of the manuscript in Line 673.

L.587; The authors should use multiplication signs, not an alphabet.

R: We apologize for this mistake. We have modified the sign “X” to “×” in Line 704.

L221-227; Fig5C, H, I are not described.

R: We apologize for this mistake. We have added the description of Fig. 5c, h, and i in Lines 277-283.

Fig3G; The authors should explain how to quantify these bacteria in more detail.

R: We appreciate this detailed suggestion. We have added the quantitation methods of these bacteria in Lines 586-590 of the Methods section.

Method

Please describe the method of LEfSe.

R: We appreciate this detailed suggestion. LEfSe (linear discriminant analysis effect size), also known as LDA Effect Size analysis, first uses the non-parametric Kruskal-Wallis’s rank sum to detect species with significant differences in abundance between different groups. Then, the Wilcoxon rank-sum is used to test the consistency of differences between different subgroups of

different species in the previous step. Finally, LDA is used to estimate the magnitude of the impact of each component (species) abundance on the different effects. We have added the method of LEfSe in the Methods section (Lines 688-696). Besides, to provide a more precise explanation of the metagenomic sequencing data analysis, we have added the part on Bioinformatics Analysis in the Methods section (Lines 680-696).

The authors should perform a variance test before examining statistical differences.

R: We appreciate this detailed suggestion. We performed the Shapiro-Wilk test to verify the normality of data. The difference between variables was evaluated by one-way analysis of variance (ANOVA) with Tukey's multiple comparisons test if data were normally distributed; otherwise, Kruskal-Wallis and Mann-Whitney U tests were used to evaluate differences. We have revised the Statistical analysis section (Lines 800-805).

L.681-682; Did the author evaluate the correlation between two bacterial abundances? If not, the authors should omit this method.

R: We appreciate this detailed suggestion. We have omitted this method since this experiment did not involve a comparison between two groups.

Please describe the clone names of the monoclonal antibodies.

R: We appreciate this detailed suggestion. We have complemented the clone names of the monoclonal antibodies used in WB experiments as the Supplementary Table 3 and the clone names of the monoclonal antibodies used in FACS experiments as the Supplementary Table 4 in the Supplemental information.

References

1. Flak, M.B., et al., *Inflammatory arthritis disrupts gut resolution mechanisms, promoting barrier breakdown by Porphyromonas gingivalis*. JCI Insight, 2019. **4**(13).
2. Flak, M.B., J.F. Neves, and R.S. Blumberg, *Welcome to the Microgenderome*. Science, 2013. **339**(6123): p. 1044-1045.
3. Hajishengallis, G., et al., *Low-abundance biofilm species orchestrates inflammatory periodontal disease through the commensal microbiota and complement*. Cell Host Microbe, 2011. **10**(5): p. 497-506.
4. Garrett, W.S., et al., *Enterobacteriaceae act in concert with the gut microbiota to induce spontaneous and maternally transmitted colitis*. Cell Host Microbe, 2010. **8**(3): p. 292-300.
5. Huang, X., et al., *Linoleic acid inhibits in vitro function of human and murine dendritic cells, CD4(+)T cells and retinal pigment epithelial cells*. Graefes Arch Clin Exp Ophthalmol, 2021. **259**(4): p. 987-998.
6. Eichele, D.D. and K.K. Kharbanda, *Dextran sodium sulfate colitis murine model: An indispensable tool for advancing our understanding of inflammatory bowel diseases pathogenesis*. World J Gastroenterol, 2017. **23**(33): p. 6016-6029.
7. Kitajima S, M.M., Sagara E, Shimizu C, Ikeda Y., *Dextran sodium sulfate-induced colitis in germ-free IQI/Jic mice*. Exp Anim., 2001. **50**(5): p. 387-95.
8. Harrington, L.E., et al., *Interleukin 17-producing CD4+ effector T cells develop via a lineage distinct from the T helper type 1 and 2 lineages*. Nat Immunol., 2005. **6**(11): p. 1123-32.
9. Stumhofer, J.S., et al., *Interleukin 27 negatively regulates the development of interleukin 17-producing T helper cells during chronic inflammation of the central nervous system*. Nat Immunol., 2006. **7**(9): p. 937-45.

10. Laurence, A., et al., *Interleukin-2 signaling via STAT5 constrains T helper 17 cell generation*. *Immunity*, 2007. **26**(3): p. 371-81.
11. Marelli, G., et al., *Non-redundant role of the chemokine receptor CX3CR1 in the anti-inflammatory function of gut macrophages*. *Immunobiology*, 2017. **222**(2): p. 463-472.
12. Kostadinova, F.I., et al., *Crucial involvement of the CX3CR1-CX3CL1 axis in dextran sulfate sodium-mediated acute colitis in mice*. *J Leukoc Biol*, 2010. **88**(1): p. 133-43.
13. Neurath, M.F., *New targets for mucosal healing and therapy in inflammatory bowel diseases*. *Mucosal Immunol*, 2014. **7**(1): p. 6-19.
14. Hegazy, A.N., et al., *Circulating and Tissue-Resident CD4(+) T Cells With Reactivity to Intestinal Microbiota Are Abundant in Healthy Individuals and Function Is Altered During Inflammation*. *Gastroenterology*, 2017. **153**(5): p. 1320-1337 e16.
15. Han, H., et al., *From gut microbiota to host appetite: gut microbiota-derived metabolites as key regulators*. *Microbiome*, 2021. **9**(1): p. 162.
16. Lavelle, A. and H. Sokol, *Gut microbiota-derived metabolites as key actors in inflammatory bowel disease*. *Nat Rev Gastroenterol Hepatol*, 2020. **17**(4): p. 223-237.
17. Nicholson, J.K., et al., *Host-Gut Microbiota Metabolic Interactions*. *Science*, 2012. **336**(6086): p. 1262-1267.
18. Yue, B., et al., *Regulation of the intestinal microbiota: An emerging therapeutic strategy for inflammatory bowel disease*. *World J Gastroenterol*, 2020. **26**(30): p. 4378-4393.
19. Kimura A, N.T., Nohara K, Fujii-Kuriyama Y, Kishimoto T., *Aryl hydrocarbon receptor regulates Stat1 activation and participates in the development of Th17 cells*. *Proc Natl Acad Sci U S A*, 2008. **105**(28): p. 9721-6.
20. Arimatsu, K., et al., *Oral pathobiont induces systemic inflammation and metabolic changes associated with alteration of gut microbiota*. *Sci Rep*, 2014. **4**: p. 4828.
21. Nakajima, M., et al., *Oral Administration of P. gingivalis Induces Dysbiosis of Gut Microbiota and Impaired Barrier Function Leading to Dissemination of Enterobacteria to the Liver*. *PLoS One*, 2015. **10**(7): p. e0134234.
22. Stojanov, S., A. Berlec, and B. Strukelj, *The Influence of Probiotics on the Firmicutes/Bacteroidetes Ratio in the Treatment of Obesity and Inflammatory Bowel disease*. *Microorganisms*, 2020. **8**(11): p. 1-12.
23. Kitamoto, S., et al., *The Intermucosal Connection between the Mouth and Gut in Commensal Pathobiont-Driven Colitis*. *Cell*, 2020. **182**(2): p. 447-462 e14.
24. Alabdulkarim, M., et al., *Alveolar bone loss in obese subjects*. *J Int Acad Periodontol*, 2005. **7**(2): p. 34-8.

REVIEWER COMMENTS

Reviewer #1 (Remarks to the Author):

The authors have performed additional experiment and made appropriate alterations to the original manuscript in order to address all my original criticisms.

Reviewer #2 (Remarks to the Author):

I have no remarks for the revised version of manuscript by Jia et al "Porphyromonas gingivalis aggravates colitis via a gut microbiota-linoleic acid 2 metabolism-Th17/Treg balance axis". The authors have properly and fully addressed all the questions raised during the first revision.

Reviewer #3 (Remarks to the Author):

The authors have addressed most of the points raised in my previous review. However, I have detected some issues that need to be fully addressed.

Experimental design

(Response Fig. 8b-c): It is necessary to quantify Pg colonizing the oral cavity of the ligature-induced periodontitis model. Additionally, the authors should cite the previous study showing the exacerbation of DSS colitis by the ligature-induced periodontitis model (DOI: <https://doi.org/10.1016/j.cell.2020.05.048>).

Result

If Response Fig. 2 would be shown as Supplemental Figure 4, the vertical axis should be displayed in a logarithmic scale.

(L. 213-217) The causal relationship of Bacteroides in exacerbating colitis has not been sufficiently demonstrated, although the correlation has been shown. The lack of a causal relationship should be described as a Limitation of the study at the end of the Discussion.

(Response-Fig.1 and Fig. 3b) Similarity between groups cannot be discussed using alpha diversity metrics. The authors should discuss it based on beta diversity. If stating that the DSS+LiPg group has lower similarity to the Control group than the DSS+DePg group, this should be addressed by visualizing unweighted Unifrac distances or similar metrics using bar plots or violin plots, for example. Also, it's not appropriate to conclude the magnitude of the difference solely based on the p-values, which represent the accuracy of statistical significance.

(Response-Fig. 5a-b) The authors mentioned that the LA concentration in the intestinal tissue was 350 μ M, but how was this calculated?

(Response Fig. 1d-f) In L. 376-377, the authors stated that "the gut microbiota exhibited decreased microbial diversity after Pg administration." These data should be included in the revised manuscript.

No response or verification has been provided for our original comments: "Does an Ahr agonist cancel LA-mediated Th17/Treg imbalance?"

Discussion

In Response Fig. 1d-f, the Simpson indexes of the DSS+LiPg group and DSS+DePg group are higher than that of the DSS+PBS group, implying higher diversity in the DSS+LiPg group. Nevertheless, the authors describe that "the DSS+LiPg group and the DSS+DePg group showed the decreased microbial diversity compared with the DSS+PBS group." This statement should be corrected.

Point-by-point Response

REVIEWER COMMENTS

Reviewer #1 (Remarks to the Author):

The authors have performed additional experiment and made appropriate alterations to the original manuscript in order to address all my original criticisms.

R: We greatly appreciate the reviewer's excellent suggestions, which have greatly improved the quality of our article.

Reviewer #2 (Remarks to the Author):

I have no remarks for the revised version of manuscript by Jia et al "Porphyromonas gingivalis aggravates colitis via a gut microbiota-linoleic acid 2 metabolism-Th17/Treg balance axis". The authors have properly and fully addressed all the questions raised during the first revision.

R: We greatly appreciate the reviewer's excellent suggestions, which have greatly improved the quality of our article.

Reviewer #3 (Remarks to the Author):

The authors have addressed most of the points raised in my previous review. However, I have detected some issues that need to be fully addressed.

R: We appreciate the reviewer's comments and suggestions. We have thoroughly considered all their points and performed additional experiments to answer their questions.

Experimental design

(Response Fig. 8b-c): It is necessary to quantify Pg colonizing the oral cavity of the ligature-induced periodontitis model. Additionally, the authors should cite the previous study showing the exacerbation of DSS colitis by the ligature-induced periodontitis model (DOI: <https://doi.org/10.1016/j.cell.2020.05.048>).

R: We appreciate the reviewer's thoughtful question and suggestion. We conducted the additional experiments to quantify Pg colonizing the oral cavity of the ligature-induced periodontitis (PD) model. Silk ligation and local infection of Pg were used to establish the PD model. We collected oral swab samples from the PD mice and the control group mice, and used qPCR to detect the absolute content of Pg in each sample. The number of Pg in the oral cavity from the PD mice was 733.5 ± 279.3 , approximately 23 times higher than the control group (31.3 ± 6.0) (2nd-Response Fig. 1). We cited the original and important study showing the exacerbation of DSS colitis by the ligature-induced periodontitis model (DOI: <https://doi.org/10.1016/j.cell.2020.05.048>) in the revised manuscript (Line 409, Reference 35).

[Figure Redacted]

2nd-Response Fig. 1. The number of Pg in the oral cavity of the ligature-induced periodontitis mice. Silk ligation and local infection of Pg to establish the periodontitis (PD) model. Oral swabs were collected from the two groups of mice. n=9. Data are presented as the mean \pm SD. ***p<0.001 by unpaired *t* test.

Result

If Response Fig. 2 would be shown as Supplemental Figure 4, the vertical axis should be displayed in a logarithmic scale.

R: We appreciate the reviewer's suggestion. We have modified and displayed the Supplemental Fig. 4b in a logarithmic scale.

(L. 213-217) The causal relationship of Bacteroidetes in exacerbating colitis has not been sufficiently demonstrated, although the correlation has been shown. The lack of a causal relationship should be described as a Limitation of the study at the end of the Discussion.

R: We appreciate the reviewer's suggestion. We have discussed the correlation between *Bacteroidetes* and colonic inflammation and added the description of the limitation of our study on the lack of the causal relationship of *Bacteroidetes* in exacerbating colitis (Lines 418-430).

(Response-Fig.1 and Fig. 3b) Similarity between groups cannot be discussed using alpha diversity metrics. The authors should discuss it based on beta diversity. If stating that the DSS+LiPg group has lower similarity to the Control group than the DSS+DePg group, this should be addressed by visualizing unweighted Unifrac distances or similar metrics using bar plots or violin plots, for example. Also, it's not appropriate to conclude the magnitude of the difference solely based on the p-values, which represent the accuracy of statistical significance.

R: We appreciate the reviewer's thoughtful question. We agree that similarity between groups cannot be discussed using alpha diversity metrics and it's not appropriate to conclude the magnitude of the difference solely based on the p-value. Actually, the PCoA analysis of the metagenomic sequence based on the Bray-Curtis distances was used to value the species' structural similarities between groups, equivalent to the beta analysis of amplicons. The closer the distance between the samples in the figure, the more similar the species composition and structure of the samples were (Fig. 3b). Here, we showed the Bray-Curtis distances data obtained through pairwise comparison of the permutational multivariate analysis of variance (PERMANOVA) (2nd-Response Table 1). The distance of the DSS+LiPg group *versus* the DSS+PBS group was 0.52 ± 0.02 , while the distance of the DSS+DePg group *versus* the DSS+PBS group was 0.36 ± 0.04 (2nd-Response Fig. 2). Thus, we concluded that the DSS+LiPg group showed lower similarity to the Control group than the DSS+DePg group. We added these data as the Supplementary Fig. 6g and modified the manuscript correspondingly (Lines 184-187).

2nd-Response Table 1. The Bray-Curtis distance matrix of PCoA analysis

[Table Redacted]

[Figure Redacted]

2nd-Response Fig. 2. The Bray-Curtis distance violin plots of PCoA analysis between groups.

(Response-Fig. 5a-b) The authors mentioned that the LA concentration in the intestinal tissue was 350 μ M, but how was this calculated?

R: We appreciate the reviewer's rigorous question. We previously detected that the average

concentration of LA in the intestine of colitis mice after gavage of Pg was about 100 µg/g using gas chromatography-mass spectrometry (GC-MS). We assumed that the density of colon was 1.02 g/cm³ (the average density of the body), and then the concentration of 100 ug/g was converted to 350 µM.

$$c(\text{LA}) = \frac{100\text{ug}}{g} = \left(\frac{0.1g}{280.45g} \cdot \text{mol}\right) / \left(\frac{1g}{1.02g} \cdot \text{cm}^3\right) \approx 350\mu\text{mol/L} = 350 \mu\text{M}$$

(Response Fig. 1d-f) In L. 376-377, the authors stated that "the gut microbiota exhibited decreased microbial diversity after Pg administration." These data should be included in the revised manuscript.

R: We appreciate the reviewer's thoughtful suggestion. We added these data as Supplementary Fig. 6a-f and modified the manuscript correspondingly (Lines 180-184).

No response or verification has been provided for our original comments: "Does an Ahr agonist cancel LA-mediated Th17/Treg imbalance?"

R: We appreciate the reviewer's thoughtful question. We conducted the flow cytometry experiments to detect whether the Ahr agonist cancels LA-mediated Th17/Treg imbalance *in vitro*. Here, we used the Ahr agonist-FICZ (6-formylindolo[3,2-b] carbazole), which had a high affinity with Ahr and caused an increase of Th17 and a decrease of Treg^{1,2}. Isolation of naïve CD4⁺ T cells and T-cell differentiation of Th17 were conducted as described in the Methods part. 50 µM LA or 300 nM FICZ³ or LA+FICZ was added to the induced Th17 cell culture system. The flow cytometry was used to detect the percentage of IL-17⁺ Th17 and Foxp3⁺ Treg in the CD4⁺ T cells in each group (2nd-Response Fig. 2). LA reduced the percentage of Th17 while promoting the Treg. On the contrary, FICZ significantly increased the Th17 but decreased the Treg. The percentage of Th17 in the LA+FICZ group (sample 4) significantly increased, and the percentage of Treg significantly decreased compared to the LA group (sample 2), indicating that FICZ canceled the LA-mediated Th17/Treg imbalance (2nd-Response Fig. 2a-c). In addition, the Th17/Treg ratio in the LA+FICZ group was significantly higher than in the control group (sample 1), indicating that the addition of FICZ reversed the negative regulatory effect of LA (2nd-Response Fig. 2d).

[Figure Redacted]

2nd-Response Fig. 3. Ahr agonist canceled LA-mediated Th17/Treg imbalance. (a) MACS-sorted naïve CD4⁺ T cells were cultured with anti-CD3/CD28 beads for three days. The expression of IL-17 and Foxp3 in naïve CD4⁺ T cells cocultured with 50 µM LA or 300 nM FICZ or LA+FICZ was detected by flow cytometry. (b-c) The proportion of IL-17⁺ Th17 and Foxp3⁺ Treg cells to total CD4⁺ T cells in each group was statistically analysed. (d) The ratio of Th17 to Treg in each group was statistically analysed. n=4. Data are presented as the mean ± SD; *p<0.05; **p<0.01; ***p<0.001 by one-way ANOVA.

Discussion

In Response Fig. 1d-f, the Simpson indexes of the DSS+LiPg group and DSS+DePg group are higher than that of the DSS+PBS group, implying higher diversity in the DSS+LiPg group. Nevertheless, the authors describe that "the DSS+LiPg group and the DSS+DePg group showed

the decreased microbial diversity compared with the DSS+PBS group." This statement should be corrected.

R: We apologize for this mistake. The Simpson indexes of the DSS+LiPg group and DSS+DePg group were higher than that of the DSS+PBS group, implying higher diversity in the DSS+LiPg group and the DSS+DePg group. We have corrected the statement and modified the manuscript correspondingly (Lines 180-184 and Line 381).

Response Reference:

- 1 Liu, X. *et al.* The role of STAT3 and AhR in the differentiation of CD4+ T cells into Th17 and Treg cells. *Medicine (Baltimore)* **96**, e6615, doi:10.1097/MD.0000000000006615 (2017).
- 2 Ehrlich, A. K., Pennington, J. M., Bisson, W. H., Kolluri, S. K. & Kerkvliet, N. I. TCDD, FICZ, and Other High Affinity AhR Ligands Dose-Dependently Determine the Fate of CD4+ T Cell Differentiation. *Toxicol Sci* **161**, 310-320, doi:10.1093/toxsci/kfx215 (2018).
- 3 Hayes, M. D., Ovcinnikovs, V., Smith, A. G., Kimber, I. & Dearman, R. J. The aryl hydrocarbon receptor: differential contribution to T helper 17 and T cytotoxic 17 cell development. *PLoS One* **9**, e106955, doi:10.1371/journal.pone.0106955 (2014).

REVIEWERS' COMMENTS

Reviewer #3 (Remarks to the Author):

The authors have adequately answered all my remarks. I congratulate them for a very interesting study.